# Vulnerability of Louisiana's coastal wetlands to present-day rates of relative sea-level rise

Krista L. Jankowski[1], Torbjörn E. Törnqvist[1] & Anjali M. Fernandes[1,†]

Coastal Louisiana has lost about 5,000 km² of wetlands over the past century and concern exists whether remaining wetlands will persist while facing some of the world's highest rates of relative sea-level rise (RSLR). Here we analyse an unprecedented data set derived from 274 rod surface-elevation table-marker horizon stations, to determine present-day surface-elevation change, vertical accretion and shallow subsidence rates. Comparison of vertical accretion rates with RSLR rates at the land surface (present-day RSLR rates are 12 ± 8 mm per year) shows that 65% of wetlands in the Mississippi Delta (SE Louisiana) may keep pace with RSLR, whereas 58% of the sites in the Chenier Plain (SW Louisiana) do not, rendering much of this area highly vulnerable to RLSR. At least 60% of the total subsidence rate occurs within the uppermost 5–10 m, which may account for the higher vulnerability of coastal Louisiana wetlands compared to their counterparts elsewhere.

[1] Department of Earth and Environmental Sciences, Tulane University, 6823 St. Charles Avenue, New Orleans, Louisiana 70118-5698, USA. † Present address: The Center for Integrative Geosciences, The University of Connecticut at Storrs, 354 Mansfield Road, Beach Hall, Storrs, Connecticut 06269, USA. Correspondence and requests for materials should be addressed to K.L.J. (email: kjankows@tulane.edu).

Coastal wetlands provide exceptionally valuable ecosystem services, including wildlife habitat, food production, biogeochemical cycling and storm-related disturbance regulation[1]. Globally, 64–71% of wetlands (including coastal wetlands) have been lost since 1900 AD (ref. 2). Louisiana is home to 40% of wetlands in the contiguous United States, yet has suffered 80% of the total wetland loss[3]. A recent analysis of surface-elevation change (SEC) and vertical accretion (VA) data from marshes in North America and Europe suggests that concerns about marsh vulnerability to sea-level rise may have been overstated[4]. However, these authors also acknowledge the 'limits to marsh adaptability in places such as coastal Louisiana', a region they recognize as underrepresented in their study. With present-day relative sea-level rise (RSLR) rates among the highest in the world (12 ± 8 mm per year), coastal Louisiana may provide a window into the future for similar settings worldwide given global sea-level predictions with similar rates later in this century.

Louisiana's coastal wetland loss ($\sim$5,000 km$^2$ between 1932 and 2010 (ref. 5)) is a complex problem, impacted by decreased sediment supply due to river leveeing and damming[6], dredging of navigation canals[7] and subsurface fluid extraction[8]. The resilience of coastal wetlands is influenced by a number of feedbacks[9,10] and for these ecosystems to persist, surface elevation must be gained at a rate that equals or exceeds that of RSLR, particularly since landward migration is not always possible[11]. Short-term (annual to decadal) resilience often involves non-linear responses to environmental factors that may be detrimental over longer timescales. For example, RSLR may induce short-term positive SEC through increased mineral deposition during inundation[12].

Modest environmental stress from RSLR may spur increased plant productivity[13], organic matter accretion and trapping of clastic sediment[10] in some cases, although for wetlands with limited elevation capital (that is, low initial elevation as is the case in coastal Louisiana) prolonged inundation decreases plant productivity[14]. Importantly, VA at the wetland surface can be partially or entirely offset by subsidence, including shallow subsidence (SS). Therefore, the interplay between these two variables determines whether net surface-elevation gain occurs[15] (Fig. 1).

An influential attempt to model marsh dynamics[13] found that salt marshes may be able to withstand RSLR rates up to $\sim$12 mm per year, given optimal primary productivity and high sediment loading in a mesotidal environment with monotonic RSLR. A subsequent synthesis of five numerical models[16] concluded that marshes with suspended sediment concentrations of $>$20 mg l$^{-1}$ and a tidal range of $<$1 m—conditions applicable to coastal Louisiana—may not become vulnerable to drowning until RSLR rates are on the order of $\sim$10 mm per year. However, observational evidence to test these model results is sparse. Currently available SEC rates are derived from relatively small case studies in coastal Louisiana[17–20] ($<$20 sites; Supplementary Table 1) and have produced results that are inconclusive as to whether RSLR outpaces wetland surface-elevation gain. In addition, the complex feedbacks of multiple variables (for example, vegetation type and initial elevation) vary among wetland sites, limiting the usefulness of data from a small number of sites to elucidate broader trends. Thus, it is increasingly clear that much larger data sets are required to address this problem[4,21].

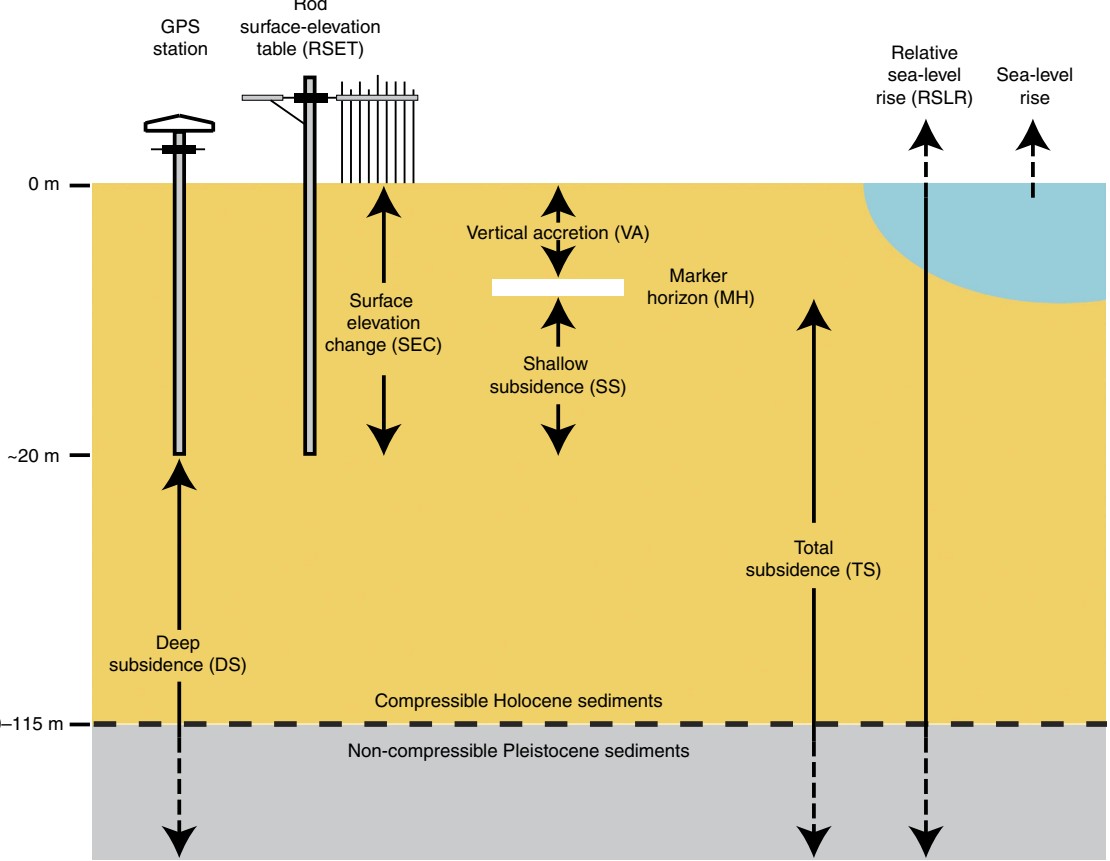

**Figure 1 | Relationship between study variables.** Representation of SEC, VA, SS (VA − SEC), DS, TS (SS + DS), sea-level rise and RSLR (TS + sea-level rise) as used in this study. When the VA rate exceeds the rate of RSLR there is an accretion surplus; when the VA rate is less than the rate of RSLR there is an accretion deficit. Figure is not to scale. Modified from ref. 22.

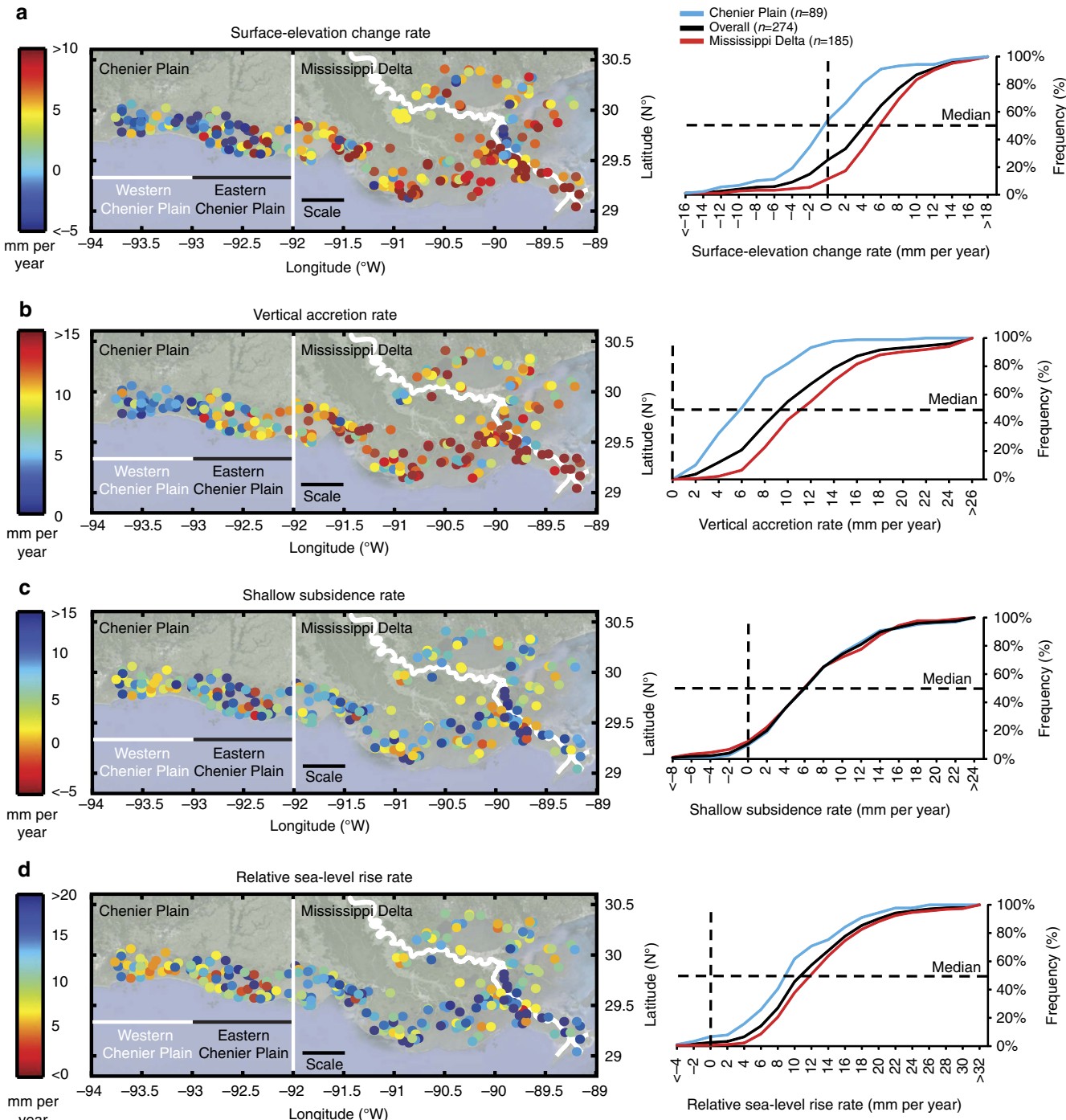

**Figure 2 | Spatial pattern and cumulative frequency distribution of present-day wetland conditions in coastal Louisiana.** Rates of (**a**) SEC; (**b**) VA; (**c**) SS; and (**d**) RSLR at 274 sites. Frequency histograms of these data sets are provided in Supplementary Fig. 1. Scale bar, 50 km.

Here we test the hypothesis that wetlands in coastal Louisiana are keeping up with present-day rates of RSLR. We evaluate 274 new records (Fig. 2) derived from rod surface-elevation table-marker horizon (RSET-MH) measurements, a well-established method for studying coastal wetland change[22]. Our data set, provided through Louisiana's Coastwide Reference Monitoring System (CRMS)[23], is an order of magnitude larger than any currently available regionally contiguous data set worldwide (Supplementary Table 1). The size and density of this data set therefore offer unprecedented opportunities for studying present-day coastal wetland dynamics along with its uncertainties, spatial

patterns and the delicate interplay between VA, SS and SEC (Fig. 1). We find that many wetland sites across coastal Louisiana, and especially those concentrated in its westernmost portion, exhibit an accretion deficit that results in vulnerability to modern rates of RSLR.

## Results

**Calculated rates.** SEC rates exhibit substantial variability among individual sites (s.d. = 7.4 mm per year; Table 1). Mean (5.7 mm per year) and median (5.8 mm per year) SEC rates are higher in the Mississippi Delta than in the Chenier Plain (−0.2 and

**Table 1 | Rates of change in coastal Louisiana wetlands.**

| Overall SEC rate (mm per year; n = 274) | | Mississippi Delta SEC rate (mm per year; n = 185) | | Chenier Plain SEC rate (mm per year; n = 89) | |
|---|---|---|---|---|---|
| Mean | 3.8 | Mean | 5.7 | Mean | − 0.2 |
| Median | 4.1 | Median | 5.8 | Median | − 0.5 |
| s.d. | 7.4 | s.d. | 7.2 | s.d. | 6.3 |
| Minimum | − 41 | Minimum | − 41 | Minimum | − 17.3 |
| Maximum | 46 | Maximum | 46 | Maximum | 22.5 |
| **Overall VA rate (mm per year; n = 274)** | | **Mississippi Delta VA rate (mm per year; n = 185)** | | **Chenier Plain VA rate (mm per year; n = 89)** | |
| Mean | 10.7 | Mean | 12.8 | Mean | 6.3 |
| Median | 9.5 | Median | 11.3 | Median | 5.9 |
| s.d. | 7.8 | s.d. | 8.4 | s.d. | 3.7 |
| Minimum | 0.2 | Minimum | 1.6 | Minimum | 0.2 |
| Maximum | 83.7 | Maximum | 83.7 | Maximum | 20.6 |
| **Overall SS rate (mm per year; n = 274)** | | **Mississippi Delta SS rate (mm per year; n = 185)** | | **Chenier Plain SS rate (mm per year; n = 89)** | |
| Mean | 6.8 | Mean | 7.1 | Mean | 6.5 |
| Median | 6 | Median | 6 | Median | 5.8 |
| s.d. | 7.9 | s.d. | 8.7 | s.d. | 6.3 |
| Minimum | − 39.4 | Minimum | − 39.4 | Minimum | − 11.2 |
| Maximum | 60.7 | Maximum | 60.7 | Maximum | 22.9 |
| **Overall TS rate (mm per year; n = 274)** | | **Mississippi Delta TS rate (mm per year; n = 185)** | | **Chenier Plain TS rate (mm per year; n = 89)** | |
| Mean | 10 | Mean | 11.2 | Mean | 7.5 |
| Median | 8.7 | Median | 10 | Median | 6.8 |
| s.d. | 8.3 | s.d. | 8.8 | s.d. | 6.3 |
| Minimum | − 35 | Minimum | − 35 | Minimum | − 10.2 |
| Maximum | 65.8 | Maximum | 65.8 | Maximum | 23.9 |
| **Overall RSLR rate (mm per year; n = 274)** | | **Mississippi Delta RSLR rate (mm per year; n = 185)** | | **Chenier Plain RSLR rate (mm per year; n = 89)** | |
| Mean | 12 | Mean | 13.2 | Mean | 9.5 |
| Median | 10.7 | Median | 12 | Median | 8.8 |
| s.d. | 8.3 | s.d. | 8.8 | s.d. | 6.3 |
| Minimum | − 33 | Minimum | − 33 | Minimum | − 8.2 |
| Maximum | 67.8 | Maximum | 67.8 | Maximum | 25.9 |

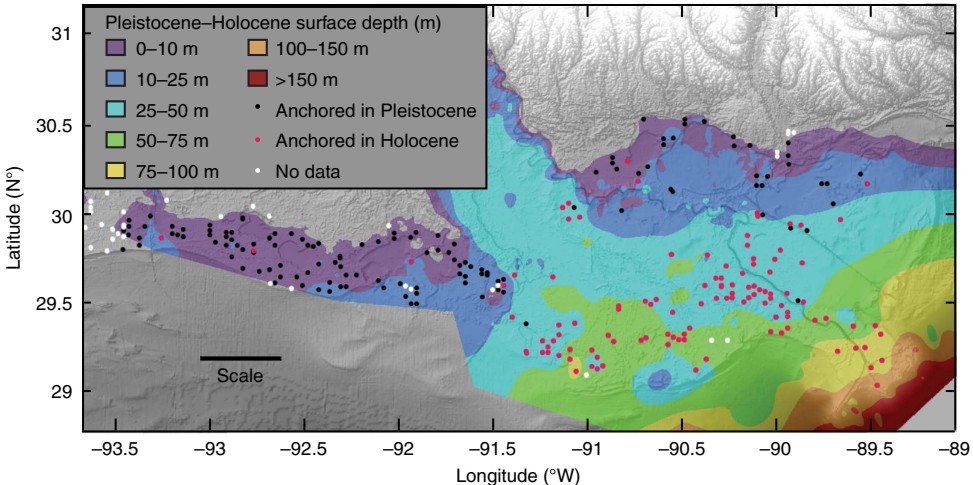

**Figure 3 | Surface-elevation table rod depth compared to the depth of the Pleistocene-Holocene contact in coastal Louisiana.** Depths are with respect to North American Vertical Datum of 1988. Because the surface elevation at the 274 sites used in this study ranges from − 0.2 to 0.6 m, the map[43] can be interpreted as a Holocene isopach map. Scale bar, 50 km.

− 0.5 mm per year, respectively) (Fig. 2a, Table 1). This difference is statistically significant according to a one-way analysis of variance (ANOVA) test ($F(1,272) = 43.5$, $P = 2.2E\text{-}10$). It is important to note that SEC rates are expressed with respect to the base of the RSET rod, which may itself be subsiding. This would make our measured rates upper limits for the true SEC values. However, recent studies[24–26] allow us to estimate deeper subsidence rates for sites anchored below the Pleistocene surface where rates are comparatively stable, as discussed in more detail below.

VA rates also vary widely (s.d. = 7.8 mm per year; Table 1). Mean (12.8 mm per year) and median (11.3 mm per year) VA rates in the Mississippi Delta are considerably higher than in the Chenier Plain (6.3 and 5.9 mm per year, respectively; Fig. 2b, Table 1); this difference is also statistically significant ($F(1,272) = 48.3$, $P = 2.7$E-10). In contrast with previous studies[10], our results do not show a significantly higher VA rate for wetlands at lower elevations as compared to those at higher elevations. This is likely due to the narrow elevation range among the sites (− 0.2 to 0.6 m, Supplementary Data 1), a reflection of the microtidal regime in coastal Louisiana. No meaningful differences in SEC and VA rates between wetland types were observed (Supplementary Table 2).

## Discussion

Differences in SEC and VA rates between the Mississippi Delta and the Chenier Plain (Supplementary Fig. 1) may be related to the differing geomorphological features and processes in each region (for example, proximity to riverine sediment inputs, connectivity to the Gulf of Mexico, impact of chenier ridges and impoundments). The trend of SEC and VA rates within the

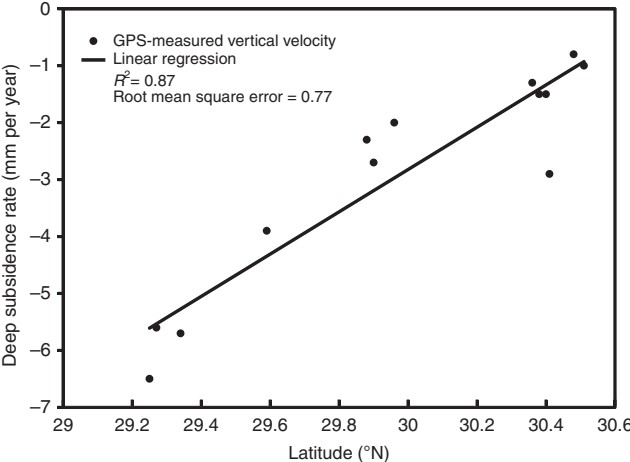

**Figure 4 | DS rates in the Mississippi Delta.** Linear regression of vertical velocity at 13 GPS stations[26] (see Supplementary Table 3) versus latitude within the Mississippi Delta. For each of the 185 Mississippi Delta sites, the DS rate was estimated by solving the linear model equation as a function of latitude.

Chenier Plain, decreasing from east to west (Fig. 2a,b), supports this interpretation. The areas with higher SEC and VA rates in the Mississippi Delta likely reflect deposition during frontal passage and tidal exchange[27], storm events[28] or major floods[29].

Our results show a high spatial variability of SS rates with no geographic trend (Fig. 2c). We find that median Mississippi Delta (6.0 mm per year) and Chenier Plain (5.8 mm per year) SS rates are strikingly similar (Fig. 2c, Table 1). Previous work has suggested that subsidence rates increase with the thickness of Holocene strata[26,30–32]. Holocene deposits are generally much thicker (32.8 ± 22.5 m) in the Mississippi Delta than in the Chenier Plain (5.8 ± 3.5 m; Fig. 3), yet the SS rates for the two regions are indistinguishable. This indicates that SS rates are primarily controlled by shallow sediment compaction known to vary rapidly over short distances[31], and fully captured in the 5–10 m thick Holocene veneer of the Chenier Plain. The median SS rate for Mississippi Delta sites anchored in the consolidated Pleistocene substrate (6.1 mm per year, $n = 64$, mean depth of Pleistocene surface 10.1 m; Fig. 3) is remarkably similar to the median SS rate for all Mississippi Delta sites, confirming that the majority of SS is occurring in the shallowest portion of the Holocene succession. Taking into account the limited thickness of Holocene strata in the Chenier Plain, we conclude that SS occurs mainly in the uppermost 5–10 m.

In order to assess whether wetlands are keeping pace with RSLR, the rate of RSLR must be determined with respect to the land surface. Tide gauges measure RSLR with respect to benchmarks that in settings like coastal Louisiana are typically anchored a few tens of metres below the land surface. In view of our finding that SS accounts for a large proportion of total subsidence (TS) and, thus, RSLR, tide gauges in this region do not capture the full amount of RSLR. Therefore, unlike most previous studies that have relied on tide gauges, we calculate the rate of RSLR with respect to the land surface for each individual site by adding an estimate of the deep subsidence (DS) rate and the present-day rate of sea-level rise in the Gulf of Mexico to the known SS rate. In the Mississippi Delta, RSET-MH measurements of SS and GPS measurements of DS are complementary, given that GPS anchor depths (mostly > 15 m; Supplementary Table 3) are generally comparable to mean RSET rod depths (22.9 ± 6.3 m). GPS records show that DS rates increase toward the coast[26] with an approximately linear trend[32]. We use this relationship (Fig. 4) to quantify the DS rate as a function of latitude at the Mississippi Delta sites. For Chenier Plain sites, a rate of 1 mm per year is used to characterize DS, based on GPS

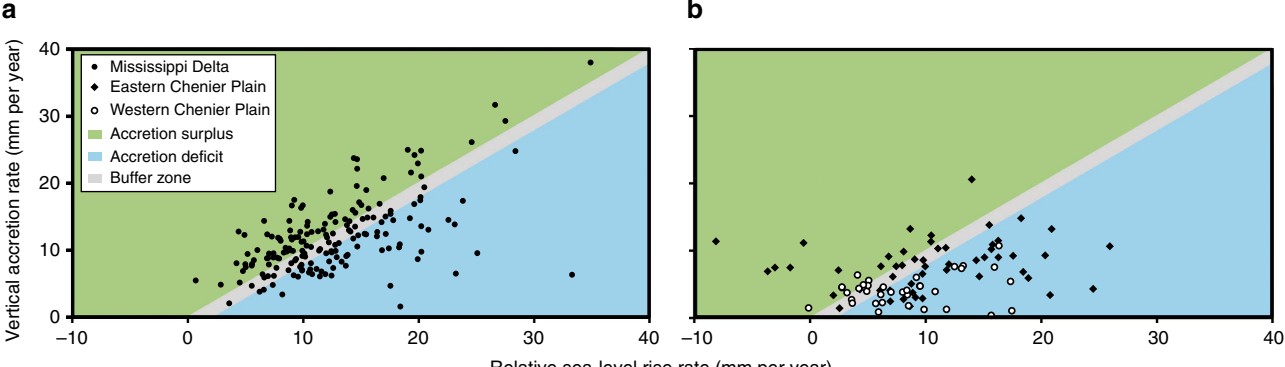

**Figure 5 | Vulnerability of coastal Louisiana wetlands to present-day rates of RSLR.** (**a**) Mississippi Delta; (**b**) Chenier Plain. Sites that fall within the accretion surplus field have VA rates that exceed the rate of RSLR. Sites within the grey buffer zone have an accretion deficit that is < 2 mm per year and are assumed herein not to be vulnerable (although this is uncertain). Sites that fall below this buffer zone have an accretion deficit > 2 mm per year and are considered vulnerable given current rates of RSLR. Outliers in the Mississippi Delta ($n = 5$, with 4 experiencing an accretion surplus) are not shown for ease of comparison between the two regions.

data from outside the Mississippi Delta[26] and supported by glacial isostatic adjustment modelling for the central US Gulf Coast[25], in turn constrained by high-resolution Holocene RSL data[24]. The sum of SS and DS yields the TS rate (Fig. 1) for each site (Table 1) and we find that SS accounts for 60 and 85% (on average) of TS in the Mississippi Delta and Chenier Plain, respectively. Finally, we add the mean rate of sea-level rise in the Gulf of Mexico from 1992–2011 satellite altimetry data ($2.0 \pm 0.4$ mm per year)[33]. The mean present-day rate of RSLR is $12.0 \pm 8.3$ mm per year overall (Fig. 2d, Table 1) and $13.2 \pm 8.8$ and $9.5 \pm 6.3$ mm per year in the Mississippi Delta and the Chenier Plain, respectively. These rates are slightly higher than previously published rates of RSLR[34] of $\sim 12$ and 6 mm per year as obtained from tide-gauge records in the two regions, respectively. It should be noted, however, that our methodology is fundamentally different. Tide gauges measure RSLR with respect to an anchor point well below the land surface and therefore do not capture the SS component. The relatively high rates that were recorded nevertheless can most likely be attributed to the fact that the published tide-gauge data cover a time interval considerably older than our records. It has been shown[8] that the high rates of oil and gas extraction between 1950 and 1980 were a likely contributor to the anomalously high rates of RSLR during that time period.

We assess present-day coastal wetland vulnerability by plotting RSLR rates versus VA rates and find that the data in the Mississippi Delta generally cluster around a 1:1 line (Fig. 5a). Sites that fall slightly below the 1:1 line may also persist, as a limited accretion deficit can be counteracted by increased productivity in the short term. Therefore, we consider sites with an accretion deficit of $>2$ mm per year below the present-day RSLR rate particularly vulnerable. In the Mississippi Delta, 35% of sites exhibit such an accretion deficit. In contrast, despite the lower rates of RSLR, the Chenier Plain (Fig. 5b) is currently facing accretion deficits at 58% of the sites. The Chenier Plain exhibits a higher concentration of vulnerable sites in its western portion, where 64% of sites are not keeping up with RSLR. These findings are consistent with the sustained wetland loss in this area[5], although it is important to note that our analysis does not specifically consider the possible impact of smaller scale geomorphological features (for example, alluvial ridges, chenier ridges) on inundation patterns. Both regions feature striking juxtapositions (Supplementary Data 1) with highly vulnerable sites that are in close proximity to sites where the VA rate currently exceeds the rate of RSLR.

We should stress that our assessment of wetland vulnerability is highly conservative. First, the methodology excludes any sites with negative VA (that is, erosion), thus inflating mean VA rates. Second, the sites are inherently biased toward stability because RSET-MH stations tend not to be located in rapidly degrading settings (for example, wetlands that are currently converting to open water). Finally, while the mean rate of sea-level rise in the Gulf of Mexico appears to have remained below the global mean over the past few decades[27], it is far from certain that this will continue to be the case in the future, considering, for example, increasing rates of Antarctic ice melt[35,36].

Our conclusions therefore represent a best-case scenario for coastal Louisiana and it remains to be seen whether observations for the past 5–10 years will translate into longer-term wetland resilience. While the inevitable increase in the rate of RSLR may be counteracted in carefully selected portions of the Mississippi Delta by means of major sediment diversions[37], the Chenier Plain is already in serious jeopardy due to its isolation from the primary sediment source (that is, the Mississippi River) and less favourable topographic conditions (that is, chenier ridges). This is particularly the case in the western Chenier Plain which

represents a highly vulnerable $\sim 3,000$ km² area where accretion deficits are currently already substantial. Since coastal Louisiana is currently experiencing rates of RSLR that will become increasingly common across the globe in the future, our findings may provide a less optimistic perspective on the fate of coastal wetlands worldwide than what recent studies have suggested.

## Methods

**Data source and selection criteria.** CRMS[23] has been operational since 2006 and at present consists of 391 sites across coastal Louisiana. At each CRMS site, SEC, VA and vegetation data (among others) are collected through a partnership of the United States Geological Survey and Louisiana's Coastal Protection and Restoration Authority; all raw data used in this study are publicly available[38]. A subset of 274 CRMS sites (Fig. 2, Table 1) was selected based on the following criteria: (1) they have never had to be re-established due to damage; (2) they have a continuous VA record from one set of MHs; and (3) the monitoring period is $\geq 6$ years (Supplementary Figs 3 and 4). Vegetation monitoring results were used to investigate possible differences between wetland types[39] (Supplementary Table 2).

**Data collection.** Data collection was overseen by United States Geological Survey and Coastal Protection and Restoration Authority staff through the CRMS programme and followed the RSET-MH methodology[40]. A rod was driven vertically into wetland sediments to the point of refusal (mean rod depth across coastal Louisiana is $20.6 \pm 6.8$ m) and secured with a PVC-pipe-bounded concrete casing at the surface[41]. During each site visit, the RSET arm was attached to the rod and in each of the four directions, 9 pins were lowered to rest on the wetland surface, for a total of 36 pin height measurements per site visit. Initial pin heights provide a baseline. SEC, measured biannually, was determined by taking the difference between site visit and baseline pin heights and then taking the mean of all 36 measurements.

VA measurements were carried out using a liquid $N_2$ cryo-coring methodology[42]. During the first 36 months after CRMS site establishment, measurements were taken every 6 months. Four measurements were taken from each of three MHs which comprise the first established VA plot set (known as 'Plot Set 1'). Measurements were then repeated every 18 months until the MHs were no longer intact and the plot set no longer yielded data[41]. All sites included in this study have complete VA records through Fall 2015. A mean of the 12 VA measurements was calculated after each site visit. SEC and VA rates were calculated using linear regression (Supplementary Fig. 2). SS is defined as VA minus SEC[12] relative to a fixed vertical reference point (that is, the rod depth), or the amount of SEC that does not result from the addition/subtraction of material at the land surface (Fig. 1).

CRMS protocols require SEC and VA data collection to be conducted from a raised boardwalk with access from the same location during each site visit to ensure that personnel do not disturb the site. Whenever possible, measurements at a given site are taken by the same individuals as previous measurements to ensure consistency and familiarity with site conditions. Finally, data are entered and all data are subsequently verified before being released to ensure data quality[41]. These steps limit potential error during data collection.

**Statistical analyses.** To examine the potential effect of including highly noisy records (that is, records from sites with significant scatter around the linear trend of SEC or VA) on our statistical analyses, we compared the results using the full data set ($n = 274$) to a data set with the highest SEC and/or VA r.m.s. error (Supplementary Fig. 2) records removed ($n = 227$). While this reduced the standard deviation in several cases, the mean and median values were essentially unchanged (Supplementary Table 4). As a result, we did not eliminate any CRMS sites from our statistical analysis due to noisiness.

Descriptive statistics were calculated for overall SEC, VA and SS rates, as well as for the Mississippi Delta and the Chenier Plain separately (Table 1). One-way ANOVA tests were carried out to determine if differences in mean SEC, VA and SS rates were statistically significant for the Mississippi Delta and Chenier Plain data subsets, as well as for different wetland types (Supplementary Table 2, Supplementary Fig. 5).

**Data availability.** The data that support the findings of this study are available from the Coastal Protection and Restoration Authority (CPRA) of Louisiana, via the Coastwide Reference Monitoring System programme and can be retrieved from the Coastal Information Management System (CIMS) database (http://cims.coastal.louisiana.gov).

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

## Acknowledgements

This work was supported by the US National Science Foundation (EAR-1349311), the US Department of Energy (through the National Institute for Climatic Change Research Coastal Center) and The Water Institute of the Gulf. Fieldwork was conducted and data provided through the Coastwide Reference Monitoring System. We are grateful for support, discussions and comments from a number of colleagues including Sarai Piazza, Camille Stagg and Brett Patton (US Geological Survey); Leigh Anne Sharpe and Tommy McGinnis (Coastal Protection and Restoration Authority); Mead Allison and Brendan Yuill (The Water Institute of the Gulf); Matt Kirwan (Virginia Institute of Marine Science); Paul Heinrich (Louisiana Geological Survey); John Day (Louisiana State University); Glenn Milne (University of Ottawa); and Reda Amer, Lael Vetter, and members of the Quaternary Research Group at Tulane University. The paper benefited greatly from constructive feedback by three reviewers.

## Author contributions

K.L.J. and T.E.T. conceived of and organized the study and wrote the paper; K.L.J. analysed the data and created figures; A.M.F. wrote Matlab code and contributed to spatial analyses and figures.

## Additional information

**Competing interests:** The authors declare no competing financial interests.

