## [Peer Review File · Nature Communications]

Reviewers' comments:

Reviewer #1 (Remarks to the Author):

A. Summary of key results.

The manuscript reports on a significant assessment of new data from a Surface Elevation Table array in the Mississippi delta areas of Louisiana. The key finding is that marsh wetlands in the eastern Mississippi Delta generally kept pace with RSLR, while 58% of sites in the Chenier Plain immediately to the west exhibited substantial accretion deficits. The manuscript reports a new method of determining RSLR site by site, which gives a more site specific assessment of this important variable relative to the widely used method of using regional rates.

B. Originality and interest.

I am not expert on Louisiana or the US, but my searches and plagiarism checks indicate that reports from the dataset are previously unpublished. The manuscript is in its introduction largely in reaction to a recent paper in Nature Climate Change -citation number 4- Kirwan et al. (2016), and the manuscript provides a better data set than the measurements used by Kirwan et al. (2016), being at least 5 years long and a larger dataset relative to 3 years. The manuscript also has I expect a density of measurements that is previously unreported, in the number of SET sites measured relative to spatial area. If some, a comment on this could further raise the importance of this manuscript.

The Editor asks for feedback on "will it influence thinking" in the email I got after I accepted the review. This work potentially has the best ever data set on RSET results as comparatively shown by Table S4. The article however seems a reactive response to Kirwan et al. (2010), and could attract far more interest, and influence thinking in more clearly publicising the risk of the IPCC5 sea level rise projections for marshes (and mangroves) as more of a stand-alone report. Louisiana owing to the high RSLR on that coast can show the world what the future of coastal wetlands may be like in terms of wetland loss, and so it could influence thinking by policy makers if the manuscript is less reactive and more independently reporting the study. Perhaps re-title it something like "Louisiana marshes show that sea level rise projections will be catastrophic for global coastal wetlands", and start the first sentences something like "Owing to rapid coastal subsidence, Louisiana can provide an actual example of how global coastal wetlands will respond to the sea level rise rates they will be experiencing in future decades. Here we report an extensive assessment of surface elevation change ...".

C. Data & methodology: validity of approach, quality of data, quality of presentation

The manuscript states that Kirwan et al. (2016) did not include records from coastal Louisiana. The Kirwan et al. (2016) article is rather obscure about where their sites were located, but do mention 'the Gulf' which is later indicated could be the Gulf of Mexico. As Louisiana abuts the Gulf of Mexico, confirmation of evidence in Kirwan et al. (2016) that no Louisiana sites are used would confirm the author's statement. Kirwan et al. (2016) is also obscure on their methods, referring to "combining, refining and adding to three previous compilations", the first two of which just judging by the authors (who have published on Louisiana), could include Louisiana sites. I could not obtain these book chapters in the timeframe of doing this review to check on this exclusion of Louisiana by Kirwan et al. (2016).

The SET array that provides the results is described as set up by citation number 18, which has no authors or institutions/ affiliations in common with the manuscript's authors. The methods section describes that the data was collected by staff at other institutions, some staff of which are thanked for support and discussions in the Acknowledgements but no mention of fieldwork. A sentence describing how the data and project later became the property of staff at Tulane University would be helpful for readers of the paper to understand how the long-term project evolved.

Page 3, the sentence "A recent synthesis of five numerical models⁴ concluded that marshes with

suspended sediment concentrations of >20 mg/L and a tidal range of <1 m - conditions applicable to coastal Louisiana - may reach a tipping point (i.e., drown) at RSLR rates on the order of ~10 mm/yr" seems to be incorrect, in that the statement I reckon the manuscript authors are referring to in Kirwan et al. (2016) reads " The dynamic marsh models highlight that marshes in estuaries with very low tidal range (<1 m) and suspended sediment concentrations (<20 mg l⁻¹) will be vulnerable to middle-of-the-road IPCC sea level scenarios", hence the "> " may be round the wrong way, and "tipping point" not what was meant, as vulnerable has more ranges of meaning such as likelihood to be influenced. I expect that Louisiana waters close to the Mississippi are somewhat turbid (although I am not expert on that) so it may mean that this statement in Kirwan et al. (2016) may not in fact be applicable to coastal Louisiana as stated.

The next sentence "However, observational evidence to test these model results is sparse" needs sources and evidence or some explanation, or better linkage with the next sentence and Suppl Table 4. The logic here is broken by the lead word to the next sentence "Additionally.." implies a new subject .

Figure 1 could be more convincing. Cumulative frequency is an unclear way of graphing the rate change results data, for example the RSLR graph shows that 100% have >32 mm/ yr RSLR whereas I expect what it should show that 100% have at least 0 change. It would be more convincing if the actual results were plotted such as in a histogram.

Also on Figure 1, the legend on the top of the graphs implies that data to the left is Chenier Plain sites and data to the right is Mississippi delta sites as on the maps to the left of the graphs, hence better place the Legend in a vertical array under the title LEGEND. The colour bar between the graphs to the left and the maps to the right is unclear as to which it helps interpret, if related to the dots on the map then better place to the left of the map.

Table 1: This data would be more convincing if presented in a graph/s with bars for the standard deviation, minimum and maximum. It would enable the reader to better assess the results, and compare and contrast these different rates.

Figure 3 could be improved. The x and y axis scale are different on both so hard to compare, would be more convincing if MD and CP data were plotted on one graph, with different colours for both and a legend that makes these clear. There is a lot of text in the figure heading that explains what could be (deleted if) stated more concisely in an inserted legend of colours used. The x and y axes need marks on the axis as to where the number specifically applies. The fonts are of all sizes and all bold, be more considerate to readers in having all the same font such as the axis number font looks suitable and use regular more as is more readable than bold. For consideration of space in the journal and readability, the legend could be in a blank spot of the graph such as inserted top left corner, moving "accretion surplus " that is currently there down.

D. Appropriate use of statistics and treatment of uncertainties

The methodology description including the statistics used is the weakest part of the article, but fortunately the SET technique is well published already hence brings its own credibility. The word "campaign" is used, which is unclear, also in Figure S4. The last paragraph of the article (in the methods description) is unconvincing, and associated Supplemental Table S3 and Fig. S3 are unclear in their relevance, giving a weak endpoint to the manuscript. The discussion of error is vague- clarify the error concerns. Treatment of "highly noisy records" and "remove" and "omit" are also vaguely mentioned for it seems the first time at the end of the manuscript, this needs careful explaining such as what does "noise" mean here, also the data omission procedures for sites that did not fit the analysis, as the last sentence states that some data was omitted.

In calculation of RSLR, the authors use "mean rate of sea-level rise in the Gulf of Mexico from 1992-2011 satellite altimetry data (2.0 {plus minus} 0.4 mm/yr) ", which is likely over a different time period to the CRMS data from the SET array. This potential source of error could be better

mentioned and discussed.

E. Conclusions: robustness, validity, reliability

Kirwan et al. (2016) comment in conclusion that "Observations of marsh deterioration and conversion to open water indicate that there are limits to marsh adaptability in places such as coastal Louisiana", so contradicting their title. The manuscript could better go on to exemplify this with the new Louisiana data set.

F. Suggested improvements:

mm/ yr is not a standard unit, better to use mm yr⁻¹ (with the -1 in superscript) as in Kirwan et al. (2016).

Second paragraph, 1st sentence, is this area of loss described in the exact same location as the study area? Please clarify.

Last paragraph: RMS is an unexplained acronym. Please spell it out.

While recognising the brevity necessary for publications in the Nature Group of journals, this study would be better published without the use of so many acronyms which make it difficult to easily read and understand.

G. References: appropriate credit to previous work?

Some multi-author references have all names, which others just have "et al.". Consistency would be an improvement. My preference would be to have all authors named on all references.

H. Clarity and context: lucidity of abstract/summary, appropriateness of abstract, introduction and conclusions

3rd sentence: Reference 3 is rather an older source, could better use a more recent data to update this point.

3rd sentence needs a source in referring to "resilience", which has a literature for its use based on Holling (1973).

Line 13 and elsewhere, "unsustainable" is a misleading word in this context, and its use by Lovelock et al. (2015) and citing this word to Nicholls and Cazeanave (2010) and Woodruff et al. (2013) is incorrectly cited as it is not used by those authors. Better the manuscript authors use "vulnerable", as in the similar SET analysis by Lovelock et al. (2015). Sustainable is a word with so many different applications, better would be more direct language, such as end of the send last sentence of the first paragraph, better to delete "sustainable" and replace by "will be lost".

The term "coastwide" is unclear, please define it. Also explain how Mississippi delta sites are different from Chenier plain sites, what are the criteria. Figure 1 indicate 92 degrees longitude, but Figure 2 shows this to be a rather unremarkable longitude to pick. Figure 3 finds that -93 degrees longitude is a significant change, perhaps shift the boundary between Mississippi delta sites and Chenier plain sites to there.

Reviewer #2 (Remarks to the Author):

The paper presents an exhaustive analysis of accretion rates in the Louisiana salt marshes with a comparison to a dataset on deep subsidence. This is an incredible dataset, including more than 200 stations and well surpasses any previous study of the area. Only the dataset would deserve publication in a top journal as Nature communication. The methodology is robust and the results are very exciting, partly shifting the attention from the Mississippi delta to the Chenier area. The topic is extremely important for Louisiana and for coastal salt marshes in general, since the salt marshes in the Mississippi delta represent a large fraction of the total marsh area of North America. I therefore enthusiastically support the publication of the manuscript in Nature Communications. I have a series of questions and comments that could improve the quality of the manuscript:

1) The authors should describe upfront the quantitative links among all measured quantities, may be with a figure in the supplemental material showing how they are linked (with a cartoon of a GPS station, SET, Holocene thickness, etc., as in "A global standard for monitoring coastal wetland vulnerability to accelerated sea-level rise", Webb et al.2013). Is there an equation linking surface elevation change (SEC) to surface accretion (SA) and shallow subsidence (SS)? Can we write $SEC=SA-SS$? Is there a statistical difference between SEC and SA? Is this difference related (correlated) to SS? Is SS by definition the subsidence measured by the RSET? This should be indicated. I would also clearly indicate that Total subsidence = Shallow subsidence + Deep subsidence. It is difficult to understand how these quantities are related by reading only the text.

2) Line 115 "we assess present-day coastal wetland resilience by plotting RSLR vs. Vertical Accretion." Here there is a problem. Vertical accretion tends to overestimate an increase in surface elevation because the deposited sediment is softer and less consolidated than older sediment. This new sediment will undergo compaction and therefore reduce its thickness in a relatively short time. This very shallow compaction (in the top few tens of cm) might reduce the elevation building effect of the deposited sediment. I would suggest to compare RSLR to surface elevation change measured by SET, or at least determine whether SS is comparable to SEC in areas where subsidence is low (see also supplemental material in Kirwan et al. 2016 for a discussion). Basically there is a term not accounted here, which is the long-term compaction of the accreted sediment (few tens of cms). How would you account for that?

3) Also in Fig. 3 it is important to distinguish between low marsh and high marsh sites. Ecogeomorphic feedbacks can in fact increase plant productivity at lower elevations, so that low marshes can accrete faster (see Fagherazzi et al 2012). More importantly, at lower elevations the hydroperiod increases, and therefore there is more time for sediment to deposit on the marsh surface. Only low marsh data can actually indicate whether a marsh is surviving or not, as argued in Kirwan et al. (2016). The authors partly acknowledge this ecogeomorphic feedback, by increasing the threshold for vulnerability to a deficit above 2 mm/yr (line 119). How was this threshold chosen? It seems arbitrary to me. The analysis of Kirwan et al 2016 indicates a difference of 3.9 mm/yr between low and high marsh. I would use local data to quantify how much faster a low marsh could accrete with respect to a high marsh. For example, the authors could divide the dataset in two by absolute elevation, and check whether the marshes in low areas accrete differently from marshes in high areas.

4) One strong point of the paper is the computation of RSLR using SS and DS. I would try to compare these results to available data from tide gauges, to show that they are robust.

Minor points:

Line 37 another non-linear factor that could affect the measurement of salt marsh accretion rate is the distance from a creek. Mariotti et al 2015 (soil creep in salt marshes, geology) show that high sedimentation rates on channel banks do not necessarily reflect the accretion of the entire marsh, may be it is worthy to mention this result in the introduction.

Line 138 do you see macroscopic evidence of marsh deterioration in the Chenier area?

Line 138 I would also indicate in a sentence that marsh vulnerability in the Chenier area is likely due to the chenier ridges acting as barriers for the marsh in between them, so that those marshes are not connected to the gulf and do not receive sediment. I doubt that there is a lack of fine sediment along the entire Chenier coast for marsh accretion (after all that is a very muddy coast, without beaches, sediment concentrations in the water column are very high with respect to other areas of the US).

Reviewer #3 (Remarks to the Author):

A. Summary of key results: This article uses a large database of surface elevation and vertical accretion measurements from the Coastwide Reference Monitoring System (CRMS) in Louisiana to assess the resilience of Louisiana's coastal wetlands to present day relative sea level rise. The authors identify that the Cheniers Plains exhibit substantial accretion deficits and identified this area as unsustainable, with much of the underlying loss of resilience attributed to shallow subsidence.

B. Originality and interest: This is the first paper I have read that uses such a large dataset from the CRMS, and as the authors state is the largest dataset from RSET data to be published. This is great to see given the investment in this monitoring network. I found many of the approaches in the manuscript to be new approaches, particularly with regards to the efficacy of temporal data, and the integration of RSET data with deep subsidence measurements to determine relative sea level rise. This is indeed a new and novel approach to applying RSET data to an important question

C. Data and Methodology: The authors have evidently attempted to address many of the concerns regarding RSET data and analysis approaches, and to this end I was particularly pleased to see the spatial pattern of changes addressed in Figure 1, description of how trends were established in Figure S1, and analysis of the eventual and campaign rates of surface elevation change in Figures S3 and S4. Figures S3 and S4 provide a very visual way of assessing whether SET trends represent longer-term patterns of change or shorter term-perturbations, I have not seen this approach before. Figure S4 highlights the effect that short-term perturbations can have on surface elevation trends over shorter-time periods, in this case less than 5 years. I would be interested to see if this 5 year rate of recovery is a pattern that occurs with future perturbations, time will tell. However, it is worth noting that some of the sites in Figure S4 may also be fluctuating past the trend lines, rather than asymptoting around the eventual surface elevation change rate (e.g. CRMS0605, CRMS0355). I am aware of concerns (though have no personal experience with this) regarding the reproducibility of RSET data from such a large network of RSETs that use many different people to take measurements, I think the approach employed goes some way towards identifying RSETs with reasonable data quality. Some text indicating this would be helpful and address any further criticisms.

D. Appropriate use of statistics and treatments of uncertainty: Supplementary information page 5: This section focusses on temporal scale issues. The authors use a linear model to represent SEC trend, and this is standard practice despite the well accepted convention that vertical accretion will accelerate as sea level rise accelerates. Importantly, and worth emphasising here, the authors are comparing linear SEC trends to linear RSLR trends; presumably some of the errors in SEC trends based on linear regression are overcome by comparison with RSLR using similar approaches. In addition, I don't think it is the best justification to indicate that you use a longer analysis period than other published work (e.g. Lovelock et al. 2015). The approach the authors have taken in this paper was stringent and thoughtful; it does not require comparison with analyses using shorter periods (which may have been appropriate and was justified in this case using alternative methods).

E. Conclusions and robustness: The authors importantly place their analysis within the context of existing work by Kirwan et al. which proposes a general overestimation of the vulnerability of marshes globally. Indeed, what the authors of this study are highlighting is that spatial considerations are important when making grand statements about the vulnerability of marshes, I am glad to see such moderation of these statements. Scale is indeed important, and this manuscript may also suffer from some scale issues that I will attempt to detail below. The authors

broadly indicate that wetlands in the Mississippi Delta will generally keep pace with RSLR, while the Chenier Plains will exhibit substantial accretion deficits. While the term 'generally' has been added to generalise the outcomes for the region, there is likely to be some spatial patterning in vulnerability within the Mississippi Delta and Cheniers Plains where some sites, or clusters of sites, are more vulnerable than others; conversely some sites are more resilient than others. This should at least be acknowledged in the manuscript, as the current statements suffer from the same generalisation as Kirwan, though at a smaller scale. In addition, it would be useful for the manuscript to upscale some of the information to an international scale - can you identify other areas (probably deltas) where marsh outcomes have not be adequately considered in global analyses? Is there a reason why they are not adequately represented?

While the authors have focussed on a regional comparison of results from the Mississippi Delta region and Chenier Plain region, the authors have not given any consideration to the coastal processes in these regions that contribute to their variation in geomorphic shape, and that may also contribute to their overall resilience. There is no discussion of the potential role of Cheniers in enhancing the resilience of coastal wetlands and offsetting the accretion deficits that were observed. These cheniers may act as a coastal defence to erosion from sea-level rise, and may buffer sea level intrusion; sand and shell material requires higher energy to erode and cheniers form natural levees. It would be useful to see some discussion of the protective influence that cheniers may provide for coastal wetlands (coastal wetlands occur in this region partly due to the formation of chenier ridges), and the effect this may have in offsetting the observed accretion deficits in this region. In addition, 'elevation capital', that is the surplus elevation that a marsh has before it is converted to open water, has not been discussed in the manuscript. Analysis of elevation capital provides a generalised temporal scale from which to assess the vulnerability of sites to sea level rise.

F. Suggested improvements: I think the manuscript does not actually assess the resilience of Louisiana's coastal wetlands, rather it is an indication of the resilience, which does not include consideration elevation capital or coastal processes and geomorphic features that reduce the effects of RSLR. These factors can be readily overcome, and I encourage the authors to do this as I found the manuscript to be very enjoyable and thoughtful. Adding additional information about elevation capital is difficult, a generalised approach comparing RSET position against MSL (i.e. bath-fill approaches) will not be accurate as tidal waters are not a flat plain, as they inundate surfaces they can become attenuated or amplified. To this end I suggest altering the title of the manuscript to better match what is being achieved, and acknowledging the effect that chenier and elevation deficits may play on the resilience of coastal wetlands in the two regions. In finishing, I found this manuscript to be well-written and thoughtful. A few additional suggestions are provided below:

- Line 25: Can you define what you mean by unsustainable.
- Line 150: change subsurface to 'driven vertically into wetlands sediments'
- Supplementary information page 2: Is 'Anomalous events' the correct description. Anomalous makes it sound like they are incorrect measurements, when they may actually represent a short-term perturbation. Another description would be better
- Table S2: These wetlands classes are particularly related to wetland types in Louisiana and require further explanation for an international audience. What is the vegetation structure, species, diversity? Where are they positioned in the landscape and what geomorphic features are they associated with? How distal are they from the coastline and streams? What are the typical salinity ranges of inundating waters? An additional table providing descriptive information would be helpful

G. References: I was happy with the degree of referencing for the manuscript

H. Clarity and context: I really enjoyed reading this paper. It was well-written and thoughtful. The abstract could be shorter by removing all the ecosystem services in the first line. I would like to

see reference to the geomorphic differences between the Mississippi delta and Chenier plains, and how this may alter the resilience identified using RSETs alone.

Reviewers' original comments are included below in black text.

Manuscript Authors' responses to Reviewers' comments are included where necessary below the relevant comment in green, bold text.

Additional edits to the manuscript that were not prompted by a Reviewer comment are described in a bulleted list in blue, bold text at the end of this document.

Reviewer #1 (Remarks to the Author):

A. Summary of key results.

The manuscript reports on a significant assessment of new data from a Surface Elevation Table array in the Mississippi delta areas of Louisiana. The key finding is that marsh wetlands in the eastern Mississippi Delta generally kept pace with RSLR, while 58% of sites in the Chenier Plain immediately to the west exhibited substantial accretion deficits. The manuscript reports a new method of determining RSLR site by site, which gives a more site specific assessment of this important variable relative to the widely used method of using regional rates.

B. Originality and interest.

I am not expert on Louisiana or the US, but my searches and plagiarism checks indicate that reports from the dataset are previously unpublished. The manuscript is in its introduction largely in reaction to a recent paper in Nature Climate Change -citation number 4- Kirwan et al. (2016), and the manuscript provides a better data set than the measurements used by Kirwan et al. (2016), being at least 5 years long and a larger dataset relative to 3 years. The manuscript also has I expect a density of measurements that is previously unreported, in the number of SET sites measured relative to spatial area. If some, a comment on this could further raise the importance of this manuscript.

We have included language about the high density of RSETs (in addition to language about the large number of RSETs) in manuscript line 70: "Our data set (Supplementary Table 2), provided through Louisiana's Coastwide Reference Monitoring System (CRMS)²³, is an order of magnitude larger than any currently available regionally continuous data set worldwide (Supplementary Table 1). The size and density of this data set therefore offer unprecedented opportunities for studying present-day coastal wetland dynamics along with its uncertainties, spatial patterns, and the delicate interplay between VA, SS, and SEC (Fig. 1)."

The Editor asks for feedback on "will it influence thinking" in the email I got after I accepted the review. This work potentially has the best ever data set on RSET results as comparatively shown by Table S4. The article however seems a reactive response to Kirwan et al. (2010), and could attract far more interest, and influence thinking in more clearly publicising the risk of the IPCC5 sea level rise projections for marshes (and mangroves) as more of a stand-alone report. Louisiana owing to the high RSLR on that coast can show the world what the future of coastal wetlands may be like in terms of wetland loss, and so it could influence thinking by policy makers if the manuscript is less reactive and more independently reporting the study.

Starting with manuscript line 35, we have included language which points to the importance of modern conditions in Louisiana in terms of future global conditions along coasts: “With present-day relative sea-level rise (RSLR) rates among the highest in the world (coastwide mean of 12 mm yr⁻¹), coastal Louisiana may provide a window into the future for similar settings worldwide given sea-level predictions of similar rates later in this century.”

In our conclusion, we have also added language starting in manuscript line 160 to address this comment: “Since coastal Louisiana is already experiencing rates of RSLR that will become increasingly common across the globe in the future, our findings may provide a less optimistic perspective on the fate of coastal wetlands worldwide than what recent studies have suggested.”

Perhaps re-title it something like "Louisiana marshes show that sea level rise projections will be catastrophic for global coastal wetlands", and start the first sentences something like "Owing to rapid coastal subsidence, Louisiana can provide an actual example of how global coastal wetlands will respond to the sea level rise rates they will be experiencing in future decades. Here we report an extensive assessment of surface elevation change ...".

We have changed the title of the manuscript to “Vulnerability of Louisiana’s coastal wetlands to present-day rates of relative sea-level rise” in order to better indicate that Louisiana wetlands are vulnerable under modern conditions while being careful to not overstate the conclusions or misrepresent the framing within the manuscript. We agree that the role of Louisiana as a possible window into future conditions for coastal communities and feel that the aims of this comment are met with the above mentioned revisions, but we prefer to remain cautious about extrapolating our findings to coastal wetlands globally.

C. Data & methodology: validity of approach, quality of data, quality of presentation

The manuscript states that Kirwan et al. (2016) did not include records from coastal Louisiana. The Kirwan et al. (2016) article is rather obscure about where their sites were located, but do mention 'the Gulf' which is later indicated could be the Gulf of Mexico. As Louisiana abuts the Gulf of Mexico, confirmation of evidence in Kirwan et al. (2016) that no Louisiana sites are used would confirm the author's statement. Kirwan et al. (2016) is also obscure on their methods, referring to "combining, refining and adding to three previous compilations", the first two of which just judging by the authors (who have published on Louisiana), could include Louisiana sites. I could not obtain these book chapters in the timeframe of doing this review to check on this exclusion of Louisiana by Kirwan et al. (2016).

From Kirwan et al. (2016): ““These comparisons are simplistic, and based on haphazardly distributed studies that do not reflect the actual geographic distribution of marshes. For example, most of the marshes included in our database are from the Atlantic Coast of North America and dominated by *S. patens* or *S. alterniflora*. Although roughly 50% of Atlantic and Gulf Coast marshes in the US are located along the Gulf Coast, only about 10% of accretion estimates in our dataset are from the Gulf Coast.” In all, only 10 Louisiana sites are used in the Kirwan et al. (2016) meta-analysis and only 2 of those sites include SET measurements.

In light of this, we have altered the language to more accurately describe how Louisiana records are used in the Kirwan et al. (2016) study. Starting in manuscript line 31, we state: “A recent analysis of

surface-elevation change and vertical accretion data from marshes in North America and Europe suggests that concerns about marsh vulnerability to sea-level rise may have been overstated⁴. However, these authors also recognize the “limits to marsh adaptability in places such as coastal Louisiana”, a region they recognize as underrepresented in their study.”

The SET array that provides the results is described as set up by citation number 18, which has no authors or institutions/ affiliations in common with the manuscript's authors. The methods section describes that the data was collected by staff at other institutions, some staff of which are thanked for support and discussions in the Acknowledgements but no mention of fieldwork. A sentence describing how the data and project later became the property of staff at Tulane University would be helpful for readers of the paper to understand how the long-term project evolved.

The Coastwide Reference Monitoring System is overseen jointly by the USGS and the Coastal Protection and Restoration Authority in Louisiana. In manuscript line 70 we acknowledge that CRMS is the source of the data. We have added language starting in manuscript line 170 to point out that all CRMS data is publically available online, including adding in the URL necessary to access the data.

We have added language in the Methods section to address attribution of data collection. This can be found in manuscript line 176: “Data collection was overseen by United States Geological Survey and Coastal Protection and Restoration Authority staff through CRMS and followed the RSET-MH methodology³⁷.”

In the acknowledgments section (manuscript line 307) we have added a statement crediting fieldwork and data sourcing: “Fieldwork was conducted and data provided through the Coastwide Reference Monitoring System.”

Page 3, the sentence "A recent synthesis of five numerical models⁴ concluded that marshes with suspended sediment concentrations of >20 mg/L and a tidal range of <1 m - conditions applicable to coastal Louisiana - may reach a tipping point (i.e., drown) at RSLR rates on the order of ~10 mm/yr" seems to be incorrect, in that the statement I reckon the manuscript authors are referring to in Kirwan et al. (2016) reads " The dynamic marsh models highlight that marshes in estuaries with very low tidal range (<1 m) and suspended sediment concentrations (<20 mg l⁻¹) will be vulnerable to middle-of-the-road IPCC sea level scenarios", hence the "> " may be round the wrong way, and "tipping point" not what was meant, as vulnerable has more ranges of meaning such as likelihood to be influenced. I expect that Louisiana waters close to the Mississippi are somewhat turbid (although I am not expert on that) so it may mean that this statement in Kirwan et al. (2016) may not in fact be applicable to coastal Louisiana as stated.

In Kirwan et al. (2016) (which builds off of an earlier paper Kirwan et al. in 2010, our ref 16) the authors state that for areas with low tide (<1 m) and low suspended sediment concentrations (<20 mg l⁻¹), even middle of the road IPCC estimates will lead to drowning given their model predictions. While coastal Louisiana has a micro-tidal regime and fits the low tide (<1 m) classification, there are generally higher suspended sediment concentrations than the low figure cited by Kirwan et al., just as the Reviewer suggests.

Given this, we state that for our case with suspended sediment concentrations $>20 \text{ mg l}^{-1}$, that we are likely to have wetlands withstand sea-level rise at rates of up to $\sim 10 \text{ mm yr}^{-1}$ as shown in Figure 3a of that paper. Starting with manuscript line 56: “A subsequent synthesis of five numerical models¹⁶ concluded that marshes with suspended sediment concentrations of $>20 \text{ mg l}^{-1}$ and a tidal range of $<1 \text{ m}$ – conditions applicable to coastal Louisiana – may not become vulnerable to drowning until RSLR rates are on the order of $\sim 10 \text{ mm yr}^{-1}$.”

We have also altered the language here to use the term vulnerable rather than talking about a tipping point as suggested by the reviewer.

The next sentence "However, observational evidence to test these model results is sparse" needs sources and evidence or some explanation, or better linkage with the next sentence and Suppl Table 4. The logic here is broken by the lead word to the next sentence "Additionally.." implies a new subject .

The passage for this comment has been re-arranged for clarity. Starting from manuscript line 59: “However, observational evidence to test these model results is sparse. Currently available SEC rates are derived from relatively small case studies in coastal Louisiana^{17–20} (<20 sites; Supplementary Table 1) and have produced results that are inconclusive as to whether RSLR outpaces wetland surface-elevation gain. Additionally, the complex feedbacks of multiple variables (e.g., vegetation type and initial elevation) vary among wetland sites, limiting the usefulness of data from a few sites to elucidate larger scale trends. Thus, it is increasingly clear that much larger datasets are required to address this problem^{4,21}.”

This revision now links the idea of sparse observational evidence to small-sized case studies and inconclusive results. With this re-arrangement of thoughts, the phrase “Additionally” now does in fact change the subject, to another challenge of previous results (i.e. intersite variability). We have also added in references 17-20, which were previously only mentioned in what is now Supplementary Table 1, to the manuscript text.

Figure 1 could be more convincing. Cumulative frequency is an unclear way of graphing the rate change results data, for example the RSLR graph shows that 100% have $>32 \text{ mm/ yr}$ RSLR whereas I expect what it should show that 100% have at least 0 change. It would be more convincing if the actual results were plotted such as in a histogram.

We have included histograms in the Supplementary Information document (Supplementary Figure 1) and choose to retain the cumulative frequency curves for what is now Figure 2 in the manuscript text. In reading these curves, any point along the curve represents the percentage of all sites (y-axis) that are at or below a particular variable measurement value (x-axis) – a widely used visualization method. We have added a dashed line at 0 and at the median in the interest of clarity.

Also on Figure 1, the legend on the top of the graphs implies that data to the left is Chenier Plain sites and data to the right is Mississippi delta sites as on the maps to the left of the graphs, hence better place the Legend in a vertical array under the title LEGEND. The colour bar between the graphs to the

left and the maps to the right is unclear as to which it helps interpret, if related to the dots on the map then better place to the left of the map.

Alterations to what is now Figure 2 have been made as suggested by the Reviewer, including shifting of the color bar to the left of the maps and re-formatting of the cumulative frequency legend.

Table 1: This data would be more convincing if presented in a graph/s with bars for the standard deviation, minimum and maximum. It would enable the reader to better assess the results, and compare and contrast these different rates.

These rates (with the exception of total subsidence) are visually represented in a way that can be easily compared (coastwide vs. Mississippi Delta vs. Chenier Plain) and furthers the study narrative elsewhere in the manuscript (what is now Figure 2) or Supplementary Information (Supplementary Figure 1). We find value in the inclusion of a table for reference to our statistical analysis results, particularly for policymakers and other scientists working on related problems. By presenting these values graphically, we would force users to estimate values from graphs rather than having the actual values easily at hand.

Figure 3 could be improved. The x and y axis scale are different on both so hard to compare, would be more convincing if MD and CP data were plotted on one graph, with different colours for both and a legend that makes these clear. There is a lot of text in the figure heading that explains what could be (deleted if) stated more concisely in an inserted legend of colours used. The x and y axes need marks on the axis as to where the number specifically applies. The fonts are of all sizes and all bold, be more considerate to readers in having all the same font such as the axis number font looks suitable and use regular more as is more readable than bold. For consideration of space in the journal and readability, the legend could be in a blank spot of the graph such as inserted top left corner, moving "accretion surplus " that is currently there down.

We have adjusted what is now Figure 5 to include the legend within the bounds of panel. We have also have set the x and y axes to the same scale, which required us to remove outliers from the Mississippi Delta data (n=5, 4 of which have an accretion surplus). We refrain from plotting the Mississippi Delta and Chenier Plain data on the same graph as the result is illegible, even with various colors/shapes for data points. The font size/style has been adjusted and tick marks added to the x and y axes as suggested.

D. Appropriate use of statistics and treatment of uncertainties

The methodology description including the statistics used is the weakest part of the article, but fortunately the SET technique is well published already hence brings its own credibility.

As noted by the Reviewer, the RSET-MH methodology has been well established as a global standard for monitoring wetlands (see Webb et al., 2013) and we use previously published descriptions of the methodology as the framework for our methods section.

The word "campaign" is used, which is unclear, also in Figure S4.

We have included language in manuscript line 180 to define the term campaign: “During each site visit (referred to herein as “campaign”)...”

The last paragraph of the article (in the methods description) is unconvincing, and associated Supplemental Table S3 and Fig. S3 are unclear in their relevance, giving a weak endpoint to the manuscript.

The methods section has been rearranged to avoid a “weak” ending as much as possible. Given the format of Nature group papers to end with the methods section, we are unable to end the paper with a culminating passage rather than details on methodology. We do, however, provide a strong conclusion to the body of the paper starting from manuscript line 153: “Our conclusions therefore represent a best-case scenario for coastal Louisiana and it remains to be seen whether observations for the past 5-10 years will translate into longer-term wetland resilience. While the inevitable increase in the rate of RSLR may be counteracted in carefully selected portions of the Mississippi Delta by means of major sediment diversions³⁵, the Chenier Plain is already in serious jeopardy due to its isolation from the primary sediment source (i.e., the Mississippi River) and less favorable topographic conditions (i.e., chenier ridges). This is particularly the case in the western Chenier Plain which represents a highly vulnerable ~3000 km² area where accretion deficits are currently already substantial. Since coastal Louisiana is already experiencing rates of RSLR that will become increasingly common across the globe in the future, our findings may provide a less optimistic perspective on the fate of coastal wetlands worldwide than what recent studies have suggested.”

The discussion of error is vague- clarify the error concerns. Treatment of "highly noisy records" and "remove" and "omit" are also vaguely mentioned for it seems the first time at the end of the manuscript, this needs careful explaining such as what does "noise" mean here, also the data omission procedures for sites that did not fit the analysis, as the last sentence states that some data was omitted.

We have altered the language starting from manuscript line 195 to clarify the meaning of “highly noisy records”: “To examine the potential effect of including highly noisy records (i.e., records for sites with significant scatter around the linear trend of SEC or VA) on our statistical analyses, we compared the results using the full dataset (n = 274) to a dataset with the highest SEC and/or VA root mean square error (Supplementary Fig. 4) records removed (n = 227). While this reduced the standard deviation in several cases, the mean and median values were essentially unchanged (Supplementary Table 5). As a result, we did not omit any CRMS sites from our statistical analysis due to noisiness.” Within the context of this paragraph, the term “remove” is referring to sites being left out of a data set which is compared to the full study data set to determine the impact of SEC and/or VA records with large root mean square errors in their measurements on mean, median, etc. The term “omit” is used in the same way. We stress that we ultimately did NOT omit any data as a result of these comparisons.

There is a detailed explanation of the site selection criteria outlined starting from manuscript line 170: “A subset of 274 CRMS sites (Fig. 2) was selected based on the following criteria: (1) they have never had to be re-established due to damage; (2) they have a continuous VA record from one set of MHs;

and (3) the monitoring period is ≥ 5 years (Supplementary Figs. 2 and 3).” These are the only criteria used to omit any CRMS sites from our analysis.

In calculation of RSLR, the authors use "mean rate of sea-level rise in the Gulf of Mexico from 1992-2011 satellite altimetry data (2.0 ± 0.4 mm/yr)", which is likely over a different time period to the CRMS data from the SET array. This potential source of error could be better mentioned and discussed.

We acknowledge that the time period for satellite altimetry data is ~ 20 years while the time period for our RSET-MH data is $\sim 5-10$ years, though the error associated with this slight difference in observation windows is minor in the broader context of the study.

E. Conclusions: robustness, validity, reliability

Kirwan et al. (2016) comment in conclusion that "Observations of marsh deterioration and conversion to open water indicate that there are limits to marsh adaptability in places such as coastal Louisiana", so contradicting their title. The manuscript could better go on to exemplify this with the new Louisiana data set.

We have included language to highlight the Kirwan et al. 2016 contradiction as suggested by the Reviewer starting in manuscript line 33: “However, these authors also recognize the “limits to marsh adaptability in places such as coastal Louisiana”, a region they recognize as underrepresented in their study.”

F. Suggested improvements:

mm/yr is not a standard unit, better to use mm yr^{-1} (with the -1 in superscript) as in Kirwan et al. (2016).

All units have been standardized to a mm yr^{-1} format, both in the text and figures.

Second paragraph, 1st sentence, is this area of loss described in the exact same location as the study area? Please clarify.

The reference is describing land loss across coastal Louisiana and, given the spatial distribution of the RSET-MHs in this study, can be considered the same location as the study area.

Last paragraph: RMS is an unexplained acronym. Please spell it out.

We have spelled out root mean square error in all usages throughout the manuscript.

While recognising the brevity necessary for publications in the Nature Group of journals, this study would be better published without the use of so many acronyms which make it difficult to easily read and understand.

We have limited the number of acronyms to 9 essential acronyms within the manuscript: SEC, VA, SS, DS, TS, RSLR, RSET-MH and CRMS. The first 7 of these are now illustrated with Figure 1 in order to provide clarity on their meaning and relationship to each other. Figure 1 can now be referred to if

further clarity is needed by readers.

G. References: appropriate credit to previous work?

Some multi-author references have all names, which others just have "et al.". Consistency would be an improvement. My preference would be to have all authors named on all references.

The convention for Nature publications is for all names to be included for papers with 5 or fewer authors and for et al. to be used with more than 5 authors.

H. Clarity and context: lucidity of abstract/summary, appropriateness of abstract, introduction and conclusions

3rd sentence: Reference 3 is rather an older source, could better use a more recent data to update this point.

This is the best, widely known source for quantitative data on the proportion of US wetland loss taking place in Louisiana.

3rd sentence needs a source in referring to "resilience", which has a literature for its use based on Holling (1973).

Wherever possible we have changed the language of the manuscript to use the term "vulnerability" rather terms such as resilience or sustainability. However, the term resilience is widely used and understood to mean withstanding outside pressures/threats to sustainability, and therefore does not in our view necessitate a specific source.

Line 13 and elsewhere, "unsustainable" is a misleading word in this context, and its use by Lovelock et al. (2015) and citing this word to Nicholls and Cazeanave (2010) and Woodruff et al. (2013) is incorrectly cited as it is not used by those authors. Better the manuscript authors use "vulnerable", as in the similar SET analysis by Lovelock et al. (2015). Sustainable is a word with so many different applications, better would be more direct language, such as end of the send last sentence of the first paragraph, better to delete "sustainable" and replace by "will be lost".

The terms sustainable and unsustainable have been replaced where appropriate. This alters the manuscript language to use persistence and vulnerability instead.

The term "coastwide" is unclear, please define it.

This study discusses wetlands in coastal Louisiana, which can be broken down into the Chenier Plain and the Mississippi Delta as sub-regions. In context, we are confident that the term coastwide will be readily understood to mean the entirety of coastal Louisiana including both the Chenier Plain and the Mississippi Delta.

Also explain how Mississippi delta sites are different from Chenier plain sites, what are the criteria.

Figure 1 indicate 92 degrees longitude, but Figure 2 shows this to be a rather unremarkable longitude to

pick. Figure 3 finds that -93 degrees longitude is a significant change, perhaps shift the boundary between Mississippi delta sites and Chenier plain sites to there.

While there is no citation for a longitudinal boundary between the Chenier Plain and the Mississippi Delta, there is a broad consensus for where that boundary is approximately. We use -92 degrees as a delineation that approximates this understood geographic boundary.

Reviewer #2 (Remarks to the Author):

The paper presents an exhaustive analysis of accretion rates in the Louisiana salt marshes with a comparison to a dataset on deep subsidence. This is an incredible dataset, including more than 200 stations and well surpasses any previous study of the area. Only the dataset would deserve publication in a top journal as Nature communication. The methodology is robust and the results are very exciting, partly shifting the attention from the Mississippi delta to the Chenier area. The topic is extremely important for Louisiana and for coastal salt marshes in general, since the salt marshes in the Mississippi delta represent a large fraction of the total marsh area of North America. I therefore enthusiastically support the publication of the manuscript in Nature Communications. I have a series of questions and comments that could improve the quality of the manuscript:

1) The authors should describe upfront the quantitative links among all measured quantities, may be with a figure in the supplemental material showing how they are linked (with a cartoon of a GPS station, SET, Holocene thickness, etc., as in "A global standard for monitoring coastal wetland vulnerability to accelerated sea-level rise", Webb et al.2013). Is there an equation linking surface elevation change (SEC) to surface accretion (SA) and shallow subsidence (SS)? Can we write $SEC=SA-SS$? Is there a statistical difference between SEC and SA? Is this difference related (correlated) to SS? Is SS by definition the subsidence measured by the RSET? This should be indicated. I would also clearly indicate that Total subsidence = Shallow subsidence + Deep subsidence. It is difficult to understand how these quantities are related by reading only the text.

We have created a cartoon which is modified from the figure in Webb et al. (2013) the Reviewer is referencing. It is now included as Figure 1 in the main text. This figure also includes general equations at the top to better clarify the relationships between the multiple variables.

Because of what values are measured and what values are calculated, the generally accepted equation for relating SEC, VA, and SS is: $VA-SEC=SS$. Due to the inherent intersite variability in SEC and VA measurements, we do not identify any trends in the relationships between SEC, VA, and SS at any level below Coastwide or Chenier Plain vs. Mississippi Delta. We can say that SS is the subsidence that is measured by the RSET-MH (from the base of the rod to the land surface), whereas deep subsidence would be the subsidence measured by GPS (for everything below the base of the GPS monument).

We do state that SS is relative to the depth of the RSET-MH rod in manuscript line 192: "SS is defined as VA minus SEC^{12} relative to a fixed vertical reference point (i.e., the rod depth), or the amount of SEC that does not result from the addition/subtraction of material at the land surface (Fig.1)."

2) Line 115 "we assess present-day coastal wetland resilience by plotting RSLR vs. Vertical Accretion." Here there is a problem. Vertical accretion tends to overestimate an increase in surface elevation because the deposited sediment is softer and less consolidated than older sediment. This new sediment will undergo compaction and therefore reduce its thickness in a relatively short time. This very shallow compaction (in the top few tens of cm) might reduce the elevation building effect of the deposited sediment. I would suggest to compare RSLR to surface elevation change measured by SET, or at least determine whether SS is comparable to SEC in areas where subsidence is low (see also supplemental

material in Kirwan et al. 2016 for a discussion). Basically there is term not accounted here, which is the long-term compaction of the accreted sediment (few tens of cms). How would you account for that?

All RSET-MH based studies suffer from this particular shortcoming, as they rely on VA directly or indirectly through SEC which incorporates VA within the measurement as $SEC=VA-SS$. Given this, the suggested change of using SEC vs RSLR instead of VA vs RSLR will not avoid this particular shortcoming. To the extent possible, we have limited the impact of the overestimation of younger sediments by using records with a minimum length of observation. We use only sites with a record >5 years in length and our data set has a mean observation length of 8 years. So while there may be some overestimation of the most recent sediment deposit thickness, natural consolidation of materials below the most recent deposits will make each record more representative of the actual accreted material thickness over time.

3) Also in Fig. 3 it is important to distinguish between low marsh and high marsh sites. Ecogeomorphic feedbacks can in fact increase plant productivity at lower elevations, so that low marshes can accrete faster (see Fagherazzi et al 2012). More importantly, at lower elevations the hydroperiod increases, and therefore there is more time for sediment to deposit on the marsh surface. Only low marsh data can actually indicate whether a marsh is surviving or not, as argued in Kirwan et al. (2016). The authors partly acknowledge this ecogeomorphic feedback, by increasing the threshold for vulnerability to a deficit above 2 mm/yr (line 119). How was this threshold chosen? It seems arbitrary to me. The analysis of Kirwan et al 2016 indicates a difference of 3.9 mm/yr between low and high marsh. I would use local data to quantify how much faster a low marsh could accrete with respect to a high marsh. For example, the authors could divide the dataset in two by absolute elevation, and check whether the marshes in low areas accrete differently from marshes in high areas.

In the micro-tidal regime of coastal Louisiana (~0.3 m tidal range), all marshes are by definition low elevation marshes. Initial elevations (from site establishment ~5-10 years ago) are the only available elevation data for CRMS sites (See Supplementary Table 5). For our sites, the range of initial elevations is from -0.2 to 0.6 m, with a median of 0.18 m. Given this narrow range of elevations, there is no significant difference in VA measures between the sites with low (below the median) elevations and high (above the median) elevations according to a One-Way ANOVA ($F(1,272) = 0.11, p = 0.74$).

Starting from manuscript line 91, we have added language to highlight this: “In contrast with previous studies¹⁰, our results do not show a significantly higher VA rate for wetlands at lower elevations as compared to those at higher elevations. This is likely due to the narrow elevation range among the sites (-0.2 m to 0.6 m, Supplementary Table 2), a reflection of the microtidal regime in coastal Louisiana. Differences in SEC and VA rates between wetland types are also largely absent (Supplementary Table 3).

We have chosen the threshold of 2 mm yr⁻¹ as an improvement on cut-offs of this type in the literature. In Kirwan et al. (2016) the authors state: “Because of errors associated with elevation and accretion measurements, the dependence of sea level rates on the period of record, and a tendency for short-term marsh accretion rates to fluctuate around historical SLR rates, we considered a marsh to be in the process of submerging if its accretion deficit (elevation or accretion rate minus SLR rate)

was greater than 0.5 mm yr^{-1} .” Due to the higher RSLR in Louisiana than the majority of study areas for Kirwan et al. (2016), we increased this buffer zone to 2 mm yr^{-1} , which is both more appropriate for the setting and results in a more conservative interpretation of our vulnerability analysis in what is now Figure 5.

4) One strong point of the paper is the computation of RSLR using SS and DS. I would try to compare these results to available data from tide gages, to show that they are robust.

One of the primary contributions of this paper is our independence from tide gauge data. Tide gauge data is inherently measuring sea-level rise relative to the anchor point of the tide gauge rather than at the land surface. Our new approach is an important improvement as it gives us an RSLR value at the land surface which is more meaningful for understanding coastal response to RSLR. We highlight our choice to avoid tide gauge data starting from manuscript line 118: “Unlike most previous studies that have relied on tide gauges, we calculate the rate of RSLR with respect to the land surface for each individual site by adding an estimate of the deep subsidence (DS) rate and the present-day rate of sea-level rise in the Gulf of Mexico to the known SS rate.”

Minor points:

Line 37 another non-linear factor that could affect the measurement of salt marsh accretion rate is the distance from a creek. Mariotti et al 2015 (soil creep in salt marshes, geology) show that high sedimentation rates on channel banks do not necessarily reflect the accretion of the entire marsh, may be it is worthy to mention this result in the introduction.

We take the point that proximity to a waterway effects VA rates in light of the findings in Mariotti et al (2015). However, given the size of our data set and the inherent variability in RSET-MH placements in relation to local waterways, it is likely that any impacts would be smoothed out in our results.

Line 138 do you see macroscopic evidence of marsh deterioration in the Chenier area?

We now include language starting in manuscript line 139 to indicate wetland loss in the Chenier Plain as shown by Couvillion et al. (2011): “In contrast, despite the lower rates of RSLR, the Chenier Plain (Fig. 5b) is currently facing accretion deficits at 58% of the sites. The Chenier Plain also exhibits a high concentration of vulnerable sites in its western portion, where 64% of sites are not keeping up with RSLR. These findings are consistent with the sustained wetland loss in this area⁵.”

Line 138 I would also indicate in a sentence that marsh vulnerability in the Chenier area is likely due to the chenier ridges acting as barriers for the marsh in between them, so that those marshes are not connected to the gulf and do not receive sediment. I doubt that there is lack of fine sediment along the entire Chenier coast for marsh accretion (after all that is a very muddy coast, without beaches, sediment concentrations in the water column are very high with respect to other areas of the US).

We have included language to address this comment in manuscript line 98: “The higher SEC and VA

rates in the Mississippi Delta (Supplementary Fig. 1) are likely a result of proximity to riverine sediment inputs and more direct connections to the Gulf of Mexico, while the Chenier Plain is relatively isolated from these inputs due to chenier ridges and impoundments.”

Reviewer #3 (Remarks to the Author):

A. Summary of key results: This article uses a large database of surface elevation and vertical accretion measurements from the Coastwide Reference Monitoring System (CRMS) in Louisiana to assess the resilience of Louisiana's coastal wetlands to present day relative sea level rise. The authors identify that the Cheniers Plains exhibit substantial accretion deficits and identified this area as unsustainable, with much of the underlying loss of resilience attributed to shallow subsidence.

B. Originality and interest: This is the first paper I have read that uses such a large dataset from the CRMS, and as the authors state is the largest dataset from RSET data to be published. This is great to see given the investment in this monitoring network. I found many of the approaches in the manuscript to be new approaches, particularly with regards to the efficacy of temporal data, and the integration of RSET data with deep subsidence measurements to determine relative sea level rise. This is indeed a new and novel approach to applying RSET data to an important question

C. Data and Methodology: The authors have evidently attempted to address many of the concerns regarding RSET data and analysis approaches, and to this end I was particularly pleased to see the spatial pattern of changes addressed in Figure 1, description of how trends were established in Figure S1, and analysis of the eventual and campaign rates of surface elevation change in Figures S3 and S4. Figures S3 and S4 provide a very visual way of assessing whether SET trends represent longer-term patterns of change or shorter term-perturbations, I have not seen this approach before. Figure S4 highlights the effect that short-term perturbations can have on surface elevation trends over shorter-time periods, in this case less than 5 years. I would be interested to see if this 5 year rate of recovery is a pattern that occurs with future perturbations, time will tell. However, it is worth noting that some of the sites in Figure S4 may also be fluctuating past the trend lines, rather than asymptoting around the eventual surface elevation change rate (e.g. CRMS0605, CRMS0355).

We are also interested to see how this first-pass attempt at analyzing the effect of longer observation lengths will play out over longer time periods. And it is noted that not all of the records in Figure S4 nicely level out to approximate the eventual long-term trend, though it seems to be the case in the majority of sites.

I am aware of concerns (though have no personal experience with this) regarding the reproducibility of RSET data from such a large network of RSETs that use many different people to take measurements, I think the approach employed goes some way towards identifying RSETs with reasonable data quality. Some text indicating this would be helpful and address any further criticisms.

We are not aware of any publications which address the concern regarding reproducibility of RSET data as mentioned by the Reviewer. We acknowledge that the concern is not unreasonable, yet we are unable to find a source to refer to and therefore are unable to cite such a concern within the manuscript. Below, however, we provide information about the quality of RSET data as mentioned in the literature.

First, the RSET system is straight-forward in its set up and use as indicated in Webb et al. (2013) “Installation, maintenance and data collection require minor training, and a level of expertise already present in most of the governmental departments and non-governmental agencies with which we have interacted (Fig. 2).”

Second, Cahoon et al. (2002) reports on “major improvements” made to the system in order to increase precision and report that “Under ideal laboratory conditions, the 95% confidence limit for individual pin measurements averaged about ± 1.4 mm (range ± 0.7 to ± 1.9 mm). These modifications have resulted in a reduction of error by about 50%.” Further, as pointed out in Webb et al. (2013), the confidence intervals for the RSET represent “a figure well within the annual rate of eustatic SLR — the RSET is the only tool that can capture surface elevation change with this level of precision.” Therefore, while measurement reproducibility with the RSET-MH methodology has been raised by this Reviewer as a concern, the size of our data set and the length of observation (>5 years) at included sites goes a long way toward limiting the impact of any inherent human error introduced in the methodology.

D. Appropriate use of statistics and treatments of uncertainty: Supplementary information page 5: This section focusses on temporal scale issues. The authors use a linear model to represent SEC trend, and this is standard practice despite the well accepted convention that vertical accretion will accelerate as sea level rise accelerates. Importantly, and worth emphasising here, the authors are comparing linear SEC trends to linear RSLR trends; presumably some of the errors in SEC trends based on linear regression are overcome by comparison with RSLR using similar approaches.

We agree that our approach for determining RSLR is an improvement on other approaches (such as non-linear analyses of tide gage data) when compared with our linear VA data.

In addition, I don't think it is the best justification to indicate that you use a longer analysis period than other published work (e.g. Lovelock et al. 2015). The approach the authors have taken in this paper was stringent and thoughtful; it does not require comparison with analyses using shorter periods (which may have been appropriate and was justified in this case using alternative methods).

E. Conclusions and robustness: The authors importantly place their analysis within the context of existing work by Kirwan et al. which proposes a general overestimation of the vulnerability of marshes globally. Indeed, what the authors of this study are highlighting is that spatial considerations are important when making grand statements about the vulnerability of marshes, I am glad to see such moderation of these statements. Scale is indeed important, and this manuscript may also suffer from some scale issues that I will attempt to detail below.

The authors broadly indicate that wetlands in the Mississippi Delta will generally keep pace with RSLR, while the Chenier Plains will exhibit substantial accretion deficits. While the term 'generally' has been added to generalise the outcomes for the region, there is likely to be some spatial patterning in vulnerability within the Mississippi Delta and Cheniers Plains where some sites, or clusters of sites, are more vulnerable than others; conversely some sites are more resilient than others. This should at least

be acknowledged in the manuscript, as the current statements suffer from the same generalisation as Kirwan, though at a smaller scale.

We acknowledge the veracity of the Reviewer's comments, but feel as though any clustering or small scale trends that may exist can be clearly seen in what is now Figure 2. We added language starting in manuscript line 143 ("Notwithstanding these generalizations, in both areas highly vulnerable sites are in close proximity to sites where the VA rate currently exceeds the rate of RSLR.") to reinforce what is now Figure 5 which clearly shows that there are sites both in the Mississippi Delta and Chenier Plain that are vulnerable and that are resilient.

In addition, it would be useful for the manuscript to upscale some of the information to an international scale - can you identify other areas (probably deltas) where marsh outcomes have not been adequately considered in global analyses? Is there a reason why they are not adequately represented?

The reviewer suggestion to upscale this analysis is a worthy goal – and one that should be pursued in the future – but at this stage any attempts we would make to address this issue would amount to overgeneralizations and arm-waving, which we prefer to avoid.

While the authors have focussed on a regional comparison of results from the Mississippi Delta region and Chenier Plain region, the authors have not given any consideration to the coastal processes in these regions that contribute to their variation in geomorphic shape, and that may also contribute to their overall resilience. There is no discussion of the potential role of Cheniers in enhancing the resilience of coastal wetlands and offsetting the accretion deficits that were observed. These cheniers may act as a coastal defence to erosion from sea-level rise, and may buffer sea level intrusion; sand and shell material requires higher energy to erode and cheniers form natural levees. It would be useful to see some discussion of the protective influence that cheniers may provide for coastal wetlands (coastal wetlands occur in this region partly due to the formation of chenier ridges), and the effect this may have in offsetting the observed accretion deficits in this region.

We have added language to highlight that the Chenier Plain is and has been experiencing persistent land loss starting in manuscript line 139: "In contrast, despite the lower rates of RSLR, the Chenier Plain (Fig. 5b) is currently facing accretion deficits at 58% of the sites. The Chenier Plain also exhibits a high concentration of vulnerable sites in its western portion, where 64% of sites are not keeping up with RSLR. These findings are consistent with the sustained wetland loss in this area⁵."

Given this, we find it important to highlight the on-going role of chenier ridges acting as barriers rather than conduits to VA. We address this in manuscript line 98: "The higher SEC and VA rates in the Mississippi Delta (Supplementary Fig. 1) are likely a result of proximity to riverine sediment inputs and more direct connections to the Gulf of Mexico, while the Chenier Plain is relatively isolated from these inputs due to chenier ridges and impoundments."

In addition, 'elevation capital', that is the surplus elevation that a marsh has before it is converted to open water, has not been discussed in the manuscript. Analysis of elevation capital provides a

generalised temporal scale from which to assess the vulnerability of sites to sea level rise.

We have now included consideration of elevation capital to the manuscript, starting with manuscript line 47: “Modest environmental stress from RSLR may spur increased plant productivity¹³, organic matter accretion, and trapping of clastic sediment¹⁰ in some cases, although for wetlands with limited elevation capital (as in coastal Louisiana) prolonged inundation decreases plant productivity¹⁴.”

We have also included additional elevation data in the manuscript text starting in manuscript line 91: “In contrast with previous studies¹⁰, our results do not show a significantly higher VA rate for wetlands at lower elevations as compared to those at higher elevations. This is likely due to the narrow elevation range among the sites (-0.2 m to 0.6 m, Supplementary Table 2), a reflection of the microtidal regime in coastal Louisiana. Differences in SEC and VA rates between wetland types are also largely absent (Supplementary Table 3).”

F. Suggested improvements: I think the manuscript does not actually assess the resilience of Louisiana's coastal wetlands, rather it is an indication of the resilience, which does not include consideration elevation capital or coastal processes and geomorphic features that reduce the effects of RSLR. These factors can be readily overcome, and I encourage the authors to do this as I found the manuscript to be very enjoyable and thoughtful. Adding additional information about elevation capital is difficult, a generalised approach comparing RSET position against MSL (i.e. bath-fill approaches) will not be accurate as tidal waters are not a flat plain, as they inundate surfaces they can become attenuated or amplified. To this end I suggest altering the title of the manuscript to better match what is being achieved, and acknowledging the effect that chenier and elevation deficits may play on the resilience of coastal wetlands in the two regions.

We have changed the title of the manuscript from “Assessing...” to “Vulnerability of Louisiana’s coastal wetlands to present-day rates of relative sea-level rise” to better characterize the outcomes of the work. Concerns about elevation capital and the impact of the chenier ridges on sediment supply have been previously addressed in our responses here.

In finishing, I found this manuscript to be well-written and thoughtful. A few additional suggestions are provided below:

- Line 25: Can you define what you mean by unsustainable.

We have changed this language from unsustainable to “vulnerable to RSLR” to clarify the threat to the system.

- Line 150: change subsurface to 'driven vertically into wetlands sediments'

We have changed the language as suggested.

- Supplementary information page 2: Is 'Anomalous events' the correct description. Anomalous makes it

sound like they are incorrect measurements, when they may actually represent a short-term perturbation. Another description would be better

We have changed the language from anomalous event to short-term perturbation.

- Table S2: These wetlands classes are particularly related to wetland types in Louisiana and require further explanation for an international audience. What is the vegetation structure, species, diversity? Where are they positioned in the landscape and what geomorphic features are they associated with? How distal are they from the coastline and streams? What are the typical salinity ranges of inundating waters? An additional table providing descriptive information would be helpful

We have included a map of the wetland type distributions at our study sites to the Supplementary Information as Supplementary Figure 5. We also point readers to Visser et al. (2002) for information on wetland classifications for coastal Louisiana.

G. References: I was happy with the degree of referencing for the manuscript

H. Clarity and context: I really enjoyed reading this paper. It was well-written and thoughtful. The abstract could be shorter by removing all the ecosystem services in the first line. I would like to see reference to the geomorphic differences between the Mississippi delta and Chenier plains, and how this may alter the resilience identified using RSETs alone.

We have created distinct Abstract and Introduction sections and have moved the descriptions of ecosystem services to the introduction rather than the Abstract. The differences between the Mississippi Delta and Chenier Plain are addressed in manuscript lines 98-104 “The higher SEC and VA rates in the Mississippi Delta (Supplementary Fig. 1) are likely a result of proximity to riverine sediment inputs and more direct connections to the Gulf of Mexico, while the Chenier Plain is relatively isolated from these inputs due to chenier ridges and impoundments. The trend of SEC and VA rates within the Chenier Plain, decreasing from east to west (Figs. 2a, b), supports this interpretation. The areas with higher SEC and VA rates in the Mississippi Delta likely reflect deposition during frontal passage and tidal exchange²⁷, storm events²⁸, or major floods²⁹.”

.

Additional edits made not in response to Reviewer comments:

- **Abstract added as per Nature Communications checklist**
- **Section headings added as per Nature Communications checklist**
- **Added brief overview of results in last paragraph of Introduction section as per Nature Communications checklist**
- **Altered labeling of Supplementary Figures and Tables as per Nature Communications checklist**
- **Supplementary materials have been re-numbered and re-organized in order to appear in correct order within the manuscript text as per Nature Communications checklist**

- In manuscript line 129, we changed the percent of total subsidence from shallow subsidence to reflect use of median rates, rather than mean rates. This brings this analysis better in line with our overall philosophy to be conservative in our interpretations.
- Additional edits have been made to what is now Figure 2, including changing label placements and the addition of markers for 0 and the median in the cumulative frequency curves.
- We have moved what was previously Supplementary Figure 2 into the main text as Figure 4 to better illustrate how we determine our deep subsidence rate for the Mississippi Delta sites.

Reviewers' comments:

Reviewer #1 (Remarks to the Author):

The authors have addressed the comments of the reviewers well, and I'd recommend this for publication if a few minor points could be improved.

1. "Coastwide" may be a local or US name such as in the CRMS name, however it remains unclear for international readers from different contexts and with different ways of using language. It could mean coasts of the world for example. For the international audience, the article would have more impact if "Coastwide" were replaced by "Louisiana" or "overall" in its different usages in the manuscript.

2. I am glad to see better acknowledgement of the CRMS source of the data.

In line 176, would it be more accurate if "overseen by " were deleted and replaced by "collected"?

In Lines 169-170, rather than give the CRMS website in the main text, better follow the

recommendation of the CRMS for use of their data that I found on their website, see

<http://www.lacoast.gov/crms2/legal.aspx?type=citation>

3. Reviewer 2 suggests a figure in the Supplemental Material showing the contributions to the RSET measurements. The authors modify Figure 1 from Webb et al. 2013, and place it at Figure 1 in the main text. Nature Communications "publishes... research of interest to specialist within each field" (email to me from the Editor). Such an audience does not need another rendition of this diagram in the main text, as versions of it also appear as Fig. 1b in Lovelock et al. (2015) and Fig. 1 in Cahoon (2015). It is therefore better placed in the Supplemental Material. It could be improved by being lightened as the colours are too dark, and the font sizes increased given that there is a lot of empty space within the diagram.

4. The authors have moved what was previously Supplementary Figure 2 into the main text as Figure 4, to better illustrate how they determine the deep subsidence rate for the Mississippi Delta sites. This was not requested by the reviewers, and is a change that does not improve the paper. It was not one of the better Supplementary Figures, and I recommend that it does not appear in the main text. The second version is improved by the clearer x axis and with an internal legend, however the first version also had some better features, such as the orange line distinguishing that line from the axis colour. The detail in the figure heading of the first version was also helpful.

5. "Campaign" means a series of activities (commonly associated with a political application), and its use in referring to an individual site visit as part of a longer research project is therefore confusing. It would be clearer if replaced by a clearer description of what it was, "measurement". Such as:

line 180 delete "site visit (referred to herein as "campaign")" and replace with "measurement"

Lines 184 and 194, delete "campaign" and replace with "measurement"

Supplementary material, this replacement would also improve clarity, or where "measurement" is also part of the phrase, just delete "campaign". Where the word is used in a label in Suppl Fig 3, this phrase would be clearer as "Measured surface-elevation change".

6. Line 49 "limited elevation capital" is very unclear in what is meant. Please change to a more specific phrase.

7. Line 70: Webb et al. (2013) do not use the word resilience with respect to the method, use a better word that fits with what they wrote.

8. Reviewer 1 states: Some multi-author references have all names, which others just have "et al.". Consistency would be an improvement, and the authors reply "The convention for Nature publications is for all names to be included for papers with 5 or fewer authors and for et al. to be used with more than 5 authors." This is not followed however in the reference list, where about 16 references have just one author followed by "et al." Adherence to the Nature policy would be an improvement.

9. In the supplemental materials, spell out the acronym RMS at its first usage in the heading of Fig. S5, its use after that is fine as the acronym. This because RMS is not otherwise defined, and its similarity to CRMS may be confusing to readers.

Reviewer #2 (Remarks to the Author):

The authors have addressed most of my comments, and therefore I fully support the publication of this manuscript in Nature Communications.

However, one point needs to be addressed in detail:

The authors do not compare their estimate of RSLR with independent estimates derived from tidal gages. This comparison would clearly strengthen their results.

In a seminal paper (Relative Sea-Level Rise in Louisiana and the Gulf of Mexico: 1908-1988), Penland and Ramsey 1990 report a RSLR of ~ 1.2 cm/yr in the delta and ~ 0.60 cm/yr in the Chenier plain. The values in the delta are similar to those computed in this paper with a different methodology (1.32 cm yr, table 1) but RSLR in the Chenier is much higher than the values of Penland and Ramsey (~ 0.95 cm/yr).

Why is there such a significant difference? Note that tide gages data are very robust and the error is probably much lower than the estimate through SET and GPS (for example, here the GPS data are very sparse and were interpolated...). Please discuss this difference. May be adding error estimates would help.

Note that if we use the RSLR of Penland and Ramsey in the Chenier area most of the marshes will keep pace with SLR. Note also that the deep subsidence data in the Chenier is only inferred from a model; you do not have data there. Perhaps the data of Penland and Ramsey are correct, suggesting that many marshes in the Chenier are actually keeping pace with RSLR? I personally start to believe this simpler story: VA is higher in the delta because subsidence is higher while VA is lower in the Cheniers because subsidence is lower. All of this needs to be discussed.

Regarding the transformation of VA in long-term elevation, the authors could compare the density of the newly deposited layer to the average density of the marsh. If the density of the new layer (ρ_{VA}) is less than the density of the marsh (ρ_M) they can correct VA to a more realistic value ($VA * \rho_{VA}/\rho_M$). How much is the density of the accreting layer? The authors say that they use only sites with a record >5 yr, but they should also check whether the density of that layer has reached the same values of the top layer of the marsh.

Line 50-53 what about deep subsidence? Also deep subsidence can offset VA (as shown with your data)

Line 86 replace 'where rates are comparatively low' with 'which is comparatively stable'

Line 95 'Differences in SEC and VA rates between wetland types are also largely absent' a statistics is need here to show that there are no differences

Line 103 'frontal passage' do you mean 'frontal passage of storms'?

In Fig 1 I would put typical depths for the RSET and GPS station

In Fig. 1 it seems like VA is included in the measurement of RSLR, but in reality $RSLR = SS + DS + SLR$. I am assuming you measure RSLR without taking in account VA.

Supplemental Figure 1 Which panel refers to SEC, VA, and SS respectively?

Reviewer #3 (Remarks to the Author):

As this is the second time I am reviewing this manuscript I am going to focus my comments on the changes that have been made as part of the revision, and more specifically the changes that I have suggested (i.e. reviewer #3)

Firstly, I am generally happy with the changes that have been made on the basis of my review and others. I only have a few additional comments.

1. Regarding concerns with the reproducibility of RSET data and the veracity of measurements, my comment was based on discussion with other RSET users and non-users. I am not aware of any

source that you could cite regarding this concern. Your approach to dealing with this was fairly typical, in that you cite literature (Cahoon et al., Webb et al.) regarding confidence intervals and training; however in reality there are many parts of RSET data collection where errors could arise, meaning that the data is not representative of the true changes that have occurred. I am not familiar with the CRMS network or the data collection approach for the network, but I am confident that data from a large network was collected from multiple users, even for individual RSET benchmarks, that trampling of sites may have occurred either from those doing data collection, other people, cattle, animals etc., and that there was data entry errors, for example. It is the above issues that I am particularly referring to here, rather than the efficacy of the technique. I found the approach used to filter data to that of the best quality was very thoughtful, and suggest that you make the link between the above issues, and how you treated these issues by filtering to the highest quality data. I feel there is an opportunity to demonstrate here that even "questionable" data may be useful if it is treated appropriately – which I think you have done. I suspect queries may arise regarding the efficacy of results derived from the CRMS network, and you have an opportunity to deal with these concerns prior to publication.

2. Regarding clustering and smaller scale trends, I don't think you should be leaving it to the reader to interpret the important outcomes from Figure 2, particularly as the size and resolution of figure 2 is low, and the publication is for an international readership. Are the more vulnerable sites closer to channels or shorelines? Are they in marshes on the seaward side of chenier? Are they at high elevation sites where they may receive less inundation? There must be some general patterns that contribute to the outcomes you found. This is the information that is useful for an international readership, it is the information that they can take away and apply at their site, and it makes the manuscript useful for coastal researchers beyond Louisiana. This is what makes the paper of international importance – how does it apply elsewhere.

3. I also feel that you could give a fuller treatment of the difference in geomorphology and coastal processes on the delta and chenier plain – this too is information that applies to sites elsewhere. The processes contributing to the formation of the delta and chenier are significantly different and the influence of SLR on these processes will also differ. On top of these differences is the variable influence of human modifications on the landscapes of the chenier and delta plains. But for necessity, you have taken a standardised approach to assessing the vulnerability. You have suggested inserting the following text to deal with my query; "In contrast, despite the lower rates of RSLR, the Chenier Plain (Fig. 5b) is currently facing accretion deficits at 58% of the sites. The Chenier Plain also exhibits a high concentration of vulnerable sites in its western portion, where 64% of sites are not keeping up with RSLR. These findings are consistent with the sustained wetland loss in this area⁵." But the spatial pattern of wetland loss is different between the 2 regions due to the variable coastal processes. In addition, the pattern of wetland loss could arguably be higher in the delta region than the chenier plain (See Figure 13 in Blum and Roberts 2012 40: 655-683), which does not match your findings.

You have also suggested inserting the following text: "The higher SEC and VA rates in the Mississippi Delta (Supplementary Fig. 1) are likely a result of proximity to riverine sediment inputs and more direct connections to the Gulf of Mexico, while the Chenier Plain is relatively isolated from these inputs due to chenier ridges and impoundments.". But this text does not adequately address my concerns as you have not provided any evidence to support your statement that the chenier act as a barrier to sediment input; and while they may do so, it would also be a reasonable assumption that they would also act as a barrier to SLR as well by attenuating inundation patterns (which was the point I was trying to make).

I do not expect that you have a hydrodynamic dataset that would adequately indicate what effects SLR will have on VA and inundation patterns in chenier and delta plains, and I readily acknowledge that this is well beyond the scope of this manuscript; rather a caveat should be included with respect to your finding that the chenier are more vulnerable than the delta plain. This caveat should indicate that your assessment approach does not account for differences in landscape geomorphology on the chenier and delta plain, and that for necessity you have treated processes contributing to VA as being fairly homogenous and predictable (i.e. VA proportional to SLR and related to distance to channel). The variable landscape scale geomorphology will interact with sea-level rise to influence hydrodynamic conditions and VA; how this occurs is beyond the scope of this

study, but remains a future research direction.

4. I am pleased that you have referred to elevation capital, but you should also describe what this term means as the readership of this journal will not be familiar with the concepts.

Additional comments:

Abstract – I think it is a bit misleading to indicate that wetlands of the Mississippi Delta are generally keeping pace when you found that 38% of the measurements showed a deficit – I agree it is less than 58% but still constitutes more than a third of the measurements.

At lines 118-121 - you state 'Unlike most previous studies that have relied on tide gauges, we calculate the rate of RSLR with respect to the land surface for each individual site by adding an estimate of the deep subsidence (DS) rate and the present-day rate of sea-level rise in the Gulf of Mexico to the known SS rate'. It is worthwhile acknowledging that comparing to tide gauges, as per Cahoon et al. 2015, is currently the best way of getting a site specific indication of what water levels are doing over time. As the sea rises, it will not be a flat plain, and underlying geomorphology and hydrodynamic conditions can influence how sea level rise is propagated at a site. The techniques employed in this study are different, but not an improvement. I think there should be caution with the wording here, and perhaps some emphasis that there are more rigorous approaches that can be applied at the site scale.

At lines 131 - regarding the approach to determining RSLR; can you justify why satellite altimetry from 1992-2011 was used to indicate RSLR.

Our responses follow Reviewer remarks in bold text.

Reviewer #1 (Remarks to the Author):

The authors have addressed the comments of the reviewers well, and I'd recommend this for publication if a few minor points could be improved.

1. "Coastwide" may be a local or US name such as in the CRMS name, however it remains unclear for international readers from different contexts and with different ways of using language. It could mean coasts of the world for example. For the international audience, the article would have more impact if "Coastwide" were replaced by "Louisiana" or "overall" in its different usages in the manuscript.

We have changed the language in the manuscript to avoid usage of the term 'coastwide' except where necessary in the name Coastwide Reference Monitoring System (CRMS). We have substituted 'overall' or suitable terminology where necessary.

2. I am glad to see better acknowledgement of the CRMS source of the data.

In line 176, would it be more accurate if "overseen by" were deleted and replaced by "collected"?

In Lines 169-170, rather than give the CRMS website in the main text, better follow the recommendation of the CRMS for use of their data that I found on their website, see <http://www.lacoast.gov/crms2/legal.aspx?type=citation>

We have removed the URL in the text and replaced it with the suggested citation in line 177.

As for the question of language, changing 'overseen by' to 'collected' would be inaccurate. The CRMS program is managed by both the Coastal Protection and Restoration Authority (CPRA) and the United States Geological Survey (USGS), though their staff do not directly collect samples/take measurements in all cases. Because of the use of contractors within the CRMS program, it is most accurate to state that CPRA and USGS oversee the data collection as is currently stated in the main text line 182.

3. Reviewer 2 suggests a figure in the Supplemental Material showing the contributions to the RSET measurements. The authors modify Figure 1 from Webb et al. 2013, and place it at Figure 1 in the main text. Nature Communications "publishes... research of interest to specialist within each field" (email to me from the Editor). Such an audience does not need another rendition of this diagram in the main text, as versions of it also appear as Fig. 1b in Lovelock et al. (2015) and Fig. 1 in Cahoon (2015). It is therefore better placed in the Supplemental Material. It could be improved by being lightened as the colours are too dark, and the font sizes increased given that there is a lot of empty space within the diagram.

We have made the suggested changes to colors and font size to Figure 1.

We have chosen to keep Figure 1 in the main text. While we acknowledge that specialists in coastal and wetland geoscience will be familiar with the figures mentioned by Reviewer 1, we would point out some key differences between our figure and those precursors.

- 1. In our figure, the RSET rod does not extend to the Pleistocene sediments but rather ends somewhere in the Holocene sediment package. As only 46% of the RSET rods in this study are known to be anchored in the Pleistocene, our figure reflects the reality that in many cases there is a portion of the Holocene sediment package below the rod anchor depth for which elevation changes are not captured by the RSET.**
- 2. We include a GPS station in our figure because we use GPS measurements to determine our deep subsidence component of RSLR. Importantly, we show the GPS rod depth to be equivalent to the RSET rod depth to illustrate that (as we state in the manuscript) GPS measurements of deep subsidence (from the base of the GPS rod anchor depth and below, mostly >15 m) are generally complimentary to RSET-derived measurements of shallow subsidence (from the RSET rod anchor depth to the surface, mean value 22.9 ± 6.3 m in the Mississippi Delta). This shows that in many cases the deep subsidence measurement includes changes to both Holocene and Pleistocene sediment packages in this study.**
- 3. Our figure shows total subsidence, which is an important component of the RSLR calculation.**
- 4. Our figure does not include a tide gauge. This is important because the goal of this study is to understand what is happening at the land surface, whereas tide gauges measure changes to water level relative to benchmarks below the land surface. Our values for sea-level rise come from satellite altimetry data and therefore reflect what is happening at the surface.**

Given these differences, which reflect what is novel about our study and important to understanding our results, we feel that the Figure 1 adds value that the reader cannot get simply by referring to the mentioned figures by other authors.

4. The authors have moved what was previously Supplementary Figure 2 into the main text as Figure 4, to better illustrate how they determine the deep subsidence rate for the Mississippi Delta sites. This was not requested by the reviewers, and is a change that does not improve the paper. It was not one of the better Supplementary Figures, and I recommend that it does not appear in the main text. The second version is improved by the clearer x axis and with an internal legend, however the first version also had some better features, such as the orange line distinguishing that line from the axis colour. The detail in the figure heading of the first version was also helpful.

Our preference is to keep this figure in the main text because it constitutes an important element of the analysis. That said, if the Editor feels strongly that it belongs in the

Supplement, we are more than happy to move it back. Regardless, we have adopted the suggestions to improve the figure.

5. “Campaign” means a series of activities (commonly associated with a political application), and its use in referring to an individual site visit as part of a longer research project is therefore confusing. It would be clearer if replaced by a clearer description of what it was, “measurement”. Such as:

line 180 delete “site visit (referred to herein as “campaign”)” and replace with “measurement”

Lines 184 and 194, delete “campaign” and replace with “measurement”

Supplementary material, this replacement would also improve clarity, or where “measurement” is also part of the phrase, just delete “campaign”. Where the word is used in a label if Suppl Fig 3, this phrase would be clearer as “Measured surface-elevation change”.

We have changed the language in the manuscript to avoid use of the term ‘campaign’ and we now use ‘site visit’. However, given that at each site visit there are a number of individual measurements being taken (36 pin height measurements for SEC, for instance), we don’t feel that simply substituting the term ‘measurement’ for ‘campaign’ will provide the clarity Reviewer 1 seeks. The changes are highlighted in the manuscript text and have also been adopted in Supplementary materials and figures.

6. Line 49 “limited elevation capital” is very unclear in what is meant. Please change to a more specific phrase.

Our use of the term ‘limited elevation capital’ was originally requested by Reviewer 3 and refers to the fact that Louisiana wetland elevations are very low. This has been clarified in the parenthetical statement following the use of the term in manuscript lines 47-48.

7. Line 70: Webb et al. (2013) do not use the word resilience with respect to the method, use a better word that fits with what they wrote.

We have substituted ‘change’ for ‘resilience’ in manuscript line 68.

8. Reviewer 1 states: Some multi-author references have all names, which others just have "et al.". Consistency would be an improvement, and the authors reply “The convention for Nature publications is for all names to be included for papers with 5 or fewer authors and for et al. to be used with more than 5 authors.” This is not followed however in the reference list, where about 16 references have just one author followed by “et al.” Adherence to the Nature policy would be an improvement.

We are adhering to the journal convention of including all author names for papers with 1-5 authors and of including just the first author’s name followed by “et al.” when the total number of authors exceeds 5. We have reviewed our reference list and do not find any of our references to be in conflict with the journal’s conventions.

9. In the supplemental materials, spell out the acronym RMS at its first usage in the heading of Fig. S5, its use after that is fine as the acronym. This is because RMS is not otherwise defined, and its similarity to CRMS may be confusing to readers.

We have made this change.

Reviewer #2 (Remarks to the Author):

The authors have addressed most of my comments, and therefore I fully support the publication of this manuscript in Nature Communications.

However, one point needs to be addressed in detail:

The authors do not compare their estimate of RSLR with independent estimates derived from tidal gauges. This comparison would clearly strengthen their results.

In a seminal paper (Relative Sea-Level Rise in Louisiana and the Gulf of Mexico: 1908-1988), Penland and Ramsey 1990 report a RSLR of ~1.2 cm/yr in the delta and ~0.60 cm/yr in the Chenier plain. The values in the delta are similar to those computed in this paper with a different methodology (1.32 cm yr, table 1) but RSLR in the Chenier is much higher than the values of Penland and Ramsey (~0.95 cm/yr).

Why is there such a significant difference? Note that tide gauge data are very robust and the error is probably much lower than the estimate through SET and GPS (for example, here the GPS data are very sparse and were interpolated...). Please discuss this difference. Maybe adding error estimates would help.

We appreciate that the Reviewer raises this important point. We are very familiar with the Penland & Ramsey paper which we have frequently cited in the past. However, there are multiple reasons why we deliberately elected not to use tide-gauge records in the present study. One of the main findings of our work is that shallow subsidence in the uppermost 5-10 m dominates total subsidence and RSLR. Tide gauges measure RSL change with respect to benchmarks that are typically anchored well below the surface (often tens of meters), i.e., they do NOT record RSL change with respect to the wetland surface – the primary objective of the present paper. This readily explains why our rates of RSL rise for the Chenier Plain are higher than those from tide-gauge records. There are additional reasons why using the Penland & Ramsey data would be problematic. First, their time series end in 1988, which implies that they cover a time interval when global rates of sea-level rise were significantly slower than they are today (hence our choice to use a satellite altimetry dataset that matches our RSET time series quite well temporally). Furthermore, recent studies (e.g., Kolker et al., 2011) have suggested that rates of RSLR in coastal Louisiana were particularly high between 1960 and 1990 due to enhanced subsidence driven by oil and gas production, i.e., precisely the period that dominates the Penland & Ramsey time series. Finally, it is widely accepted that due to the large inter-annual variability, tide-gauge records similar in length with our RSET-data are simply too short to provide reliable data.

Of course, the RSET data are very noisy too (as we discuss in detail in the paper), but the point is that we have 274 of them, rather than just a handful (as would be the case with tide gauges).

These things said, we recognize that this important issue warrants more discussion and we have therefore added text to open the paragraph that previously started with “Unlike most previous studies...” (line 116). We now clarify why we circumvent tide-gauge records in the present study, mainly focusing on the issue that tide-gauge records by definition do not capture shallow subsidence.

Note that if we use the RSLR of Penland and Ramsey in the Chenier area most of the marshes will keep pace with SLR. Note also that the deep subsidence data in the Chenier is only inferred from a model; you do not have data there. Perhaps the data of Penland and Ramsey are correct, suggesting that many marshes in the Chenier are actually keeping pace with RSLR? I personally start to believe this simpler story: VA is higher in the delta because subsidence is higher while VA is lower in the Cheniers because subsidence is lower. All of this needs to be discussed.

As we point out above, adopting the Penland & Ramsey tide-gauge data for our analysis would be erroneous for multiple reasons. We also stress that deep subsidence in the Chenier Plain is well constrained; the modeling that we mentioned relies heavily on high-resolution Holocene RSL data, a point that we have now added in line 131. While there is a difference in the median total subsidence rate between the Mississippi Delta and Chenier Plain (10.0 vs. 7.5 mm/yr; Table 1), this is insufficient to explain the difference in VA rates (11.3 vs. 5.9 mm/yr; Table 1).

Regarding the transformation of VA in long-term elevation, the authors could compare the density of the newly deposited layer to the average density of the marsh. If the density of the new layer (ρ_{VA}) is less than the density of the marsh (ρ_M) they can correct VA to a more realistic value ($VA * \rho_{VA}/\rho_M$). How much is the density of the accreting layer? The authors say that they use only sites with a record >5yr, but they should also check whether the density of that layer has reached the same values of the top layer of the marsh.

This is an excellent suggestion, but unfortunately the data to do this analysis do not yet exist. The expectation is that the CRMS-program will collect such information in the foreseeable future and we agree that the suggested analysis would be a fruitful avenue.

Line 50-53 what about deep subsidence? Also deep subsidence can offset VA (as shown with your data)

We have changed the language in line 49-50 to show that VA can be offset by subsidence, including shallow subsidence.

Line 86 replace ‘where rates are comparatively low’ with ‘which is comparatively stable’

We have changed the language in line 84 by replacing ‘low’ with ‘stable.’

Line 95 ‘Differences in SEC and VA rates between wetland types are also largely absent’ a statistics is need here to show that there are no differences

We have clarified our language on this point to indicate that no meaningful differences in VA and SEC were observed between wetland types. Our treatment of the statistical data follows the literature in that we consistently present the statistics for significant results (i.e., the ANOVA results that confirm the difference in VA and SEC values for the Mississippi Delta and the Chenier Plain), and simply report when there are insignificant results (i.e., the lack of difference in SS values between the Mississippi Delta and the Chenier Plain). All data is available in Supplementary Table 3 for readers who may be interested in conducting further statistical tests relevant to other research questions.

Line 103 ‘frontal passages’ do you mean ‘frontal passage of storms’?

The correct term here is ‘frontal passage’; whether or not this is associated with storms is not the issue.

In Fig 1 I would put typical depths for the RSET and GPS station

We have revised Fig. 1 indicating the depth range of these systems at ~20 m and including a depth range for the Holocene-Pleistocene boundary. We also now make note that Fig 1 is not to scale to aid in understanding.

In Fig. 1 it seems like VA is included in the measurement of RSLR, but in reality $RSLR = SS + DS + SLR$. I am assuming you measure RSLR without taking in account VA.

As the equation at the top of the figure shows, VA must be known in order to determine SS (note that this is widely established within the SET community).

Supplemental Figure 1 Which panel refers to SEC, VA, and SS respectively?

We have added panel indications within the figure.

Reviewer #3 (Remarks to the Author):

As this is the second time I am reviewing this manuscript I am going to focus my comments on the changes that have been made as part of the revision, and more specifically the changes that I have suggested (i.e. reviewer #3)

Firstly, I am generally happy with the changes that have been made on the basis of my review

and others. I only have a few additional comments.

1. Regarding concerns with the reproducibility of RSET data and the veracity of measurements, my comment was based on discussion with other RSET users and non-users. I am not aware of any source that you could cite regarding this concern. Your approach to dealing with this was fairly typical, in that you cite literature (Cahoon et al., Webb et al.) regarding confidence intervals and training; however in reality there are many parts of RSET data collection where errors could arise, meaning that the data is not representative of the true changes that have occurred. I am not familiar with the CRMS network or the data collection approach for the network, but I am confident that data from a large network was collected from multiple users, even for individual RSET benchmarks, that trampling of sites may have occurred either from those doing data collection, other people, cattle, animals etc., and that there was data entry errors, for example. It is the above issues that I am particularly referring to here, rather than the efficacy of the technique. I found the approach used to filter data to that of the best quality was very thoughtful, and suggest that you make the link between the above issues, and how you treated these issues by filtering to the highest quality data. I feel there is an opportunity to demonstrate here that even “questionable” data may be useful if it is treated appropriately – which I think you have done. I suspect queries may arise regarding the efficacy of results derived from the CRMS network, and you have an opportunity to deal with these concerns prior to publication.

We feel that the Reviewer raises some interesting points in regards to data quality. We have added language in the Methods section (starting at line 201) to acknowledge that these issues exist while also highlighting the rigorous data quality control efforts that have been undertaken by the CRMS program.

2. Regarding clustering and smaller scale trends, I don't think you should be leaving it to the reader to interpret the important outcomes from Figure 2, particularly as the size and resolution of figure 2 is low, and the publication is for an international readership. Are the more vulnerable sites closer to channels or shorelines? Are they in marshes on the seaward side of chenier? Are they at high elevation sites where they may receive less inundation? There must be some general patterns that contribute to the outcomes you found. This is the information that is useful for an international readership, it is the information that they can take away and apply at their site, and it makes the manuscript useful for coastal researchers beyond Louisiana. This is what makes the paper of international importance – how does it apply elsewhere.

We fully recognize the richness of our dataset and the fact that there are numerous aspects that have yet to be fully explored. While the reviewer makes appropriate suggestions, the focus of our present analysis is deliberately broad. We reiterate here that the most striking spatial pattern (i.e., the contrast between the Mississippi Delta and the (western) Chenier Plain in terms of SEC and VA rates) is discussed in detail and our expectation is that this large-scale pattern will be of most interest to a broad, international readership. We also stress the fact that Supplementary Table 2 provides all the specific data that are fully available to anyone who wishes to explore finer scale patterns in detail. We point to this in the manuscript text starting in line 147.

3. I also feel that you could give a fuller treatment of the difference in geomorphology and coastal processes on the delta and chenier plain – this too is information that applies to sites elsewhere. The processes contributing to the formation of the delta and chenier are significantly different and the influence of SLR on these processes will also differ. On top of these differences is the variable influence of human modifications on the landscapes of the chenier and delta plains. But for necessity, you have taken a standardised approach to assessing the vulnerability. You have suggested inserting the following text to deal with my query; “In contrast, despite the lower rates of RSLR, the Chenier Plain (Fig. 5b) is currently facing accretion deficits at 58% of the sites. The Chenier Plain also exhibits a high concentration of vulnerable sites in its western portion, where 64% of sites are not keeping up with RSLR. These findings are consistent with the sustained wetland loss in this area⁵.” But the spatial pattern of wetland loss is different between the 2 regions due to the variable coastal processes. In addition, the pattern of wetland loss could arguably be higher in the delta region than the chenier plain (See Figure 13 in Blum and Roberts 2012 40: 655-683), which does not match your findings.

You have also suggested inserting the following text: “The higher SEC and VA rates in the Mississippi Delta (Supplementary Fig. 1) are likely a result of proximity to riverine sediment inputs and more direct connections to the Gulf of Mexico, while the Chenier Plain is relatively isolated from these inputs due to chenier ridges and impoundments.”. But this text does not adequately address my concerns as you have not provided any evidence to support your statement that the chenier act as a barrier to sediment input; and while they may do so, it would also be a reasonable assumption that they would also act as a barrier to SLR as well by attenuating inundation patterns (which was the point I was trying to make).

I do not expect that you have a hydrodynamic dataset that would adequately indicate what effects SLR will have on VA and inundation patterns in chenier and delta plains, and I readily acknowledge that this is well beyond the scope of this manuscript; rather a caveat should be included with respect to your finding that the chenier are more vulnerable than the delta plain. This caveat should indicate that your assessment approach does not account for differences in landscape geomorphology on the chenier and delta plain, and that for necessity you have treated processes contributing to VA as being fairly homogenous and predictable (i.e. VA proportional to SLR and related to distance to channel). The variable landscape scale geomorphology will interact with sea-level rise to influence hydrodynamic conditions and VA; how this occurs is beyond the scope of this study, but remains a future research direction.

We have now mentioned the differing regional geomorphic processes starting in line 96. In regards to the wetland loss map that constitutes Fig. 13 in Blum & Roberts (2012), that map refers to the period 1932-2000. This time window predates our dataset and is thus not necessarily directly comparable. Note that our citation (Couvillion et al., 2011) is a study that overlaps with the time window of our analysis and their results are therefore more applicable in this particular case.

We have added the caveat that our analysis does not consider the possible impact of smaller scale geomorphological features (e.g., alluvial ridges, chenier ridges) on inundation patterns, etc., as suggested by the reviewer starting in line 147.

4. I am pleased that you have referred to elevation capital, but you should also describe what this term means as the readership of this journal will not be familiar with the concepts.

We have included an explanation of what ‘limited elevation capital’ means in lines 47-48.

Additional comments:

Abstract – I think it is a bit misleading to indicate that wetlands of the Mississippi Delta are generally keeping pace when you found that 38% of the measurements showed a deficit – I agree it is less than 58% but still constitutes more than a third of the measurements.

The value we found for Mississippi Delta sites with an accretion deficit is actually 35%, but we have reworded things to minimize the perception that the Mississippi Delta is not in trouble starting in line 17.

At lines 118-121 - you state ‘Unlike most previous studies that have relied on tide gauges, we calculate the rate of RSLR with respect to the land surface for each individual site by adding an estimate of the deep subsidence (DS) rate and the present-day rate of sea-level rise in the Gulf of Mexico to the known SS rate’. It is worthwhile acknowledging that comparing to tide gauges, as per Cahoon et al. 2015, is currently the best way of getting a site specific indication of what water levels are doing over time. As the sea rises, it will not be a flat plain, and underlying geomorphology and hydrodynamic conditions can influence how sea level rise is propagated at a site. The techniques employed in this study are different, but not an improvement. I think there should be caution with the wording here, and perhaps some emphasis that there are more rigorous approaches that can be applied at the site scale.

As we pointed out in our response to Reviewer 2, tide gauge data are actually highly ambiguous in this setting. We have added in language in the manuscript to explain why we avoid tide gauge data.

At lines 131 - regarding the approach to determining RSLR; can you justify why satellite altimetry from 1992-2011 was used to indicate RSLR.

As discussed in our response to Reviewer 2, tide gauges (most commonly used to determine RSLR) do not capture RSLR at the surface and therefore are not suitable for our analysis. The satellite altimetry data that we are using offer the best temporal match of sea-surface height change data with our time series, something we now point out in the text.

REVIEWERS' COMMENTS:

Reviewer #2 (Remarks to the Author):

The authors addressed all my comments in the review. I therefore recommend publication of the manuscript.

May I ask the authors to cite in this manuscript the other dataset by Penland and Ramsey 1990? May they include the main results of that dataset in this paper? a RSLR of ~ 1.2 cm/yr in the delta and a RSLR of ~ 0.6 cm/yr in the Chenier area (Fig. 18 Of Penland and Ramsey). The readers need to have the full picture to better interpret the new data.

Response to Reviewer #2

REVIEWERS' COMMENTS:

Reviewer #2 (Remarks to the Author):

The authors addressed all my comments in the review. I therefore recommend publication of the manuscript.

May I ask the authors to cite in this manuscript the other dataset by Penland and Ramsey 1990? May they include the main results of that dataset in this paper? a RSLR of ~1.2 cm/yr in the delta and a RSLR of ~0.6 cm/yr in the Chenier area (Fig. 18 Of Penland and Ramsey). The readers need to have the full picture to better interpret the new data.

We have included the Penland and Ramsey (1990) data (and compared it with our findings) in the manuscript and added the paper as reference 34. The data inclusion and associated discussion can be found in the starting in manuscript line 140.